# Node-Specific Space Selection via Localized Geometric Hyperbolicity in Graph Neural Networks

**See Hian Lee**                                                                    *seehian001@e.ntu.edu.sg*
*Department of Electrical and Electronic Engineering*
*Nanyang Technological University*

**Ji Feng**                                                                          *jifeng@ntu.edu.sg*
*Department of Electrical and Electronic Engineering*
*Nanyang Technological University*

**Tay Wee Peng**                                                                    *wptay@ntu.edu.sg*
*Department of Electrical and Electronic Engineering*
*Nanyang Technological University*

**Reviewed on OpenReview:** *https://openreview.net/pdf?id=tHteJFeN1y*

## Abstract

Many graph neural networks have been developed to learn graph representations in either Euclidean or hyperbolic space, with all nodes' representations embedded in a single space. However, a graph can have hyperbolic and Euclidean geometries at different regions of the graph. Thus, it is sub-optimal to indifferently embed an entire graph into a single space. In this paper, we explore and analyze two notions of local hyperbolicity, describing the underlying local geometry: geometric (Gromov) and model-based, to determine the preferred space of embedding for each node. The two hyperbolicities' distributions are aligned using the Wasserstein metric such that the calculated geometric hyperbolicity guides the choice of the learned model hyperbolicity. As such our model Joint Space Graph Neural Network (JSGNN) can leverage both Euclidean and hyperbolic spaces during learning by allowing node-specific geometry space selection. We evaluate our model on both node classification and link prediction tasks and observe promising performance compared to baseline models.

## 1 Introduction

Graph neural networks (GNNs) are neural networks that learn from graph-structured data. Many works such as Graph Convolutional Network (GCN) (Kipf & Welling, 2016), Graph Attention Network (GAT) (Veličković et al., 2018), GraphSAGE (Hamilton et al., 2017) and their variants operate on the Euclidean space and have been applied in many areas such as recommender systems (Ying et al., 2018; Chen et al., 2022a), chemistry (Gilmer et al., 2017) and financial systems (Sawhney et al., 2021). Despite their remarkable accomplishments, their performances are still limited by the representation ability of Euclidean space. They are unable to achieve the best performance in situations when the data exhibit non-Euclidean characteristics such as scale-free, tree-like, or hierarchical structures (Yang et al., 2022).

As such, hyperbolic spaces have gained traction in research as they have been proven to better embed tree-like, hierarchical structures compared to the Euclidean geometry (Bachmann et al., 2019; Cho et al., 2019). Intuitively, encoding non-Euclidean structures such as trees in the Euclidean space would result in more considerable distortion since the number of nodes in a tree increases exponentially with the depth of the tree while the Euclidean space only grows polynomially (Zhu et al., 2020). In such cases, the hyperbolic geometry serves as an alternative to learning those structures with comparably smaller distortion as the hyperbolic space has the exponential growth property (Yang et al., 2022). As such, hyperbolic versions of

GNNs such as HGCN (Chami et al., 2019), HGNN (Liu et al., 2019), HGAT (Zhang et al., 2021a) and LGCN (Zhang et al., 2021b) have been proposed.

Nevertheless, real-world graphs are often complex. They are neither solely made up of Euclidean nor non-Euclidean structures alone but a mixture of geometrical structures. Consider a localized version of geometric hyperbolicity, a concept from geometry group theory measuring how tree-like the underlying space is for each node in the graph (refer to Section 3.1 for more details). We observe a mixture of local geometric hyperbolicity values in most of the benchmark datasets we employ for our experiments as seen in Fig. 2. This implies that the graphs contain a mixture of geometries and thus, it is not ideal to embed the graphs into a single geometry space, regardless of Euclidean or hyperbolic as it inevitably leads to undesired structural inductive biases and distortions (Yang et al., 2022).

Taking a graph containing both lattice-like and tree-like structures as an example, Fig. 1c and Fig. 1f shows that 15 of the blue-colored nodes in the tree structure are calculated to have 2-hop local geometric hyperbolicity value of zero, while 12 of the purple nodes have a value of one and the other 3 purple nodes (at the center of the lattice) have a value of two (the smaller the hyperbolicity value, the more hyperbolic). This localized metric can therefore serve as an indication during learning on which of the two spaces is more suitable to embed the respective nodes.

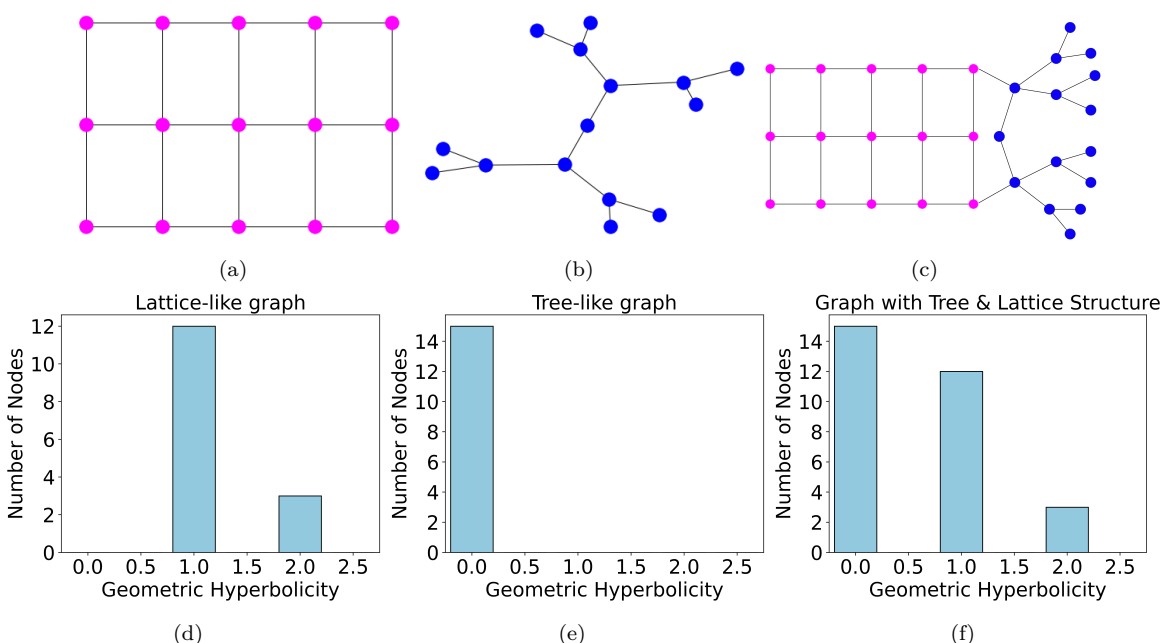

Figure 1: Example graphs. (a) Lattice-like graph. (b) A tree. (c) A combined graph containing both lattice and tree structure. (d-f) The histograms reflect the geometric hyperbolicity in the respective graphs.

Here we address this mixture of geometry in a graph and propose Joint Space Graph Neural Network (JSGNN) that performs learning on a joint space consisting of both Euclidean and hyperbolic geometries. To achieve this, we first update all the node features in both Euclidean and hyperbolic spaces independently, giving rise to two sets of updated node features. Then, we employ exponential and logarithmic maps to bridge the two spaces and an attention mechanism is used as a form of model hyperbolicity, taking into account the underlying structure around each node and the corresponding node features. The learned model hyperbolicity is guided by geometric hyperbolicity and is used to "softly decide" the most suitable embedding space for each node and to reduce the two sets of updated features into only one set. Ideally, a node should be either hyperbolic or Euclidean and not both simultaneously, thus, we also introduce an additional loss term to achieve this non-uniform characteristic. Related works are discussed in Appendix E.

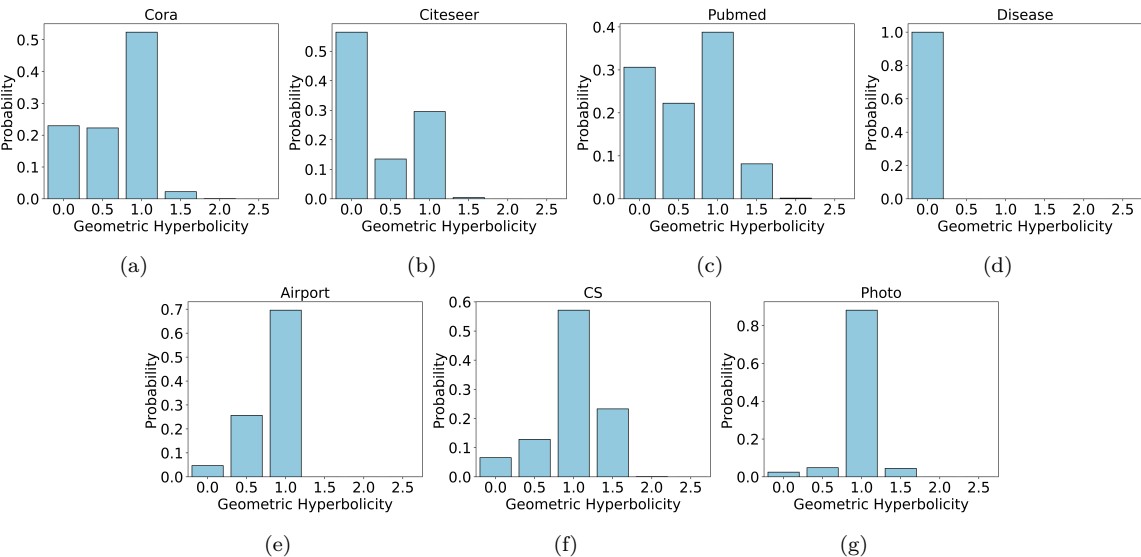

Figure 2: Distributions of geometric hyperbolicity for all datasets, obtained by computing $\delta_{G_v,\infty}$ on each nodes' 2-hop subgraph. See Appendix C for more results.

## 2 Background

In this section, we give a brief overview of hyperbolic geometry that will be used in the paper. Readers are referred to Lee (2018) for further details. Moreover, we review GAT and its hyperbolic version.

### 2.1 Hyperbolic geometry

A hyperbolic space is a non-Euclidean space with constant negative curvature. There are different but equivalent models to describe the same hyperbolic geometry. In this paper, we work with the Poincaré ball model, in which all points are inside a ball. The hyperbolic space with constant negative curvature $c$ is denoted by $(\mathbb{D}_c^n, g_{\mathbf{x}}^c)$. It consists of the $n$-dimensional hyperbolic manifold $\mathbb{D}_c^n = \{\mathbf{x} \in \mathbb{R}^n : c\|\mathbf{x}\| < 1\}$ with the Riemannian metric $g_{\mathbf{x}}^c = (\lambda_{\mathbf{x}}^c)^2 g^E$, where $\lambda_{\mathbf{x}}^c = 2/(1 - c\|\mathbf{x}\|^2)$ and $g^E = \mathbf{I}_n$ is the Euclidean metric.

At each $\mathbf{x} \in \mathbb{D}_c^n$, there is a tangent space $\mathcal{T}_{\mathbf{x}}\mathbb{D}_c^n$, which can be viewed as the first-order approximation of the hyperbolic manifold at $\mathbf{x}$ (Bachmann et al., 2019). The tangent space is then useful to perform Euclidean operations that we are familiar with but are undefined in hyperbolic spaces. A hyperbolic space and the tangent space at a point are connected through the exponential map $\exp_{\mathbf{x}}^c : \mathcal{T}_{\mathbf{x}}\mathbb{D}_c^n \to \mathbb{D}_c^n$ and logarithmic map $\log_{\mathbf{x}}^c : \mathbb{D}_c^n \to \mathcal{T}_{\mathbf{x}}\mathbb{D}_c^n$, specifically defined as follows:

$$\exp_{\mathbf{x}}^c(\mathbf{v}) = \mathbf{x} \oplus_c \left( \tanh\left( \sqrt{c}\frac{\lambda_{\mathbf{x}}^c\|\mathbf{v}\|}{2} \right) \frac{\mathbf{v}}{\sqrt{c}\|\mathbf{v}\|} \right), \tag{1}$$

$$\log_{\mathbf{x}}^c(\mathbf{y}) = \frac{2}{\sqrt{c}\lambda_{\mathbf{x}}^c} \tanh^{-1}(\sqrt{c}\| - \mathbf{x} \oplus_c \mathbf{y}\|)\frac{-\mathbf{x} \oplus_c \mathbf{y}}{\| - \mathbf{x} \oplus_c \mathbf{y}\|}, \tag{2}$$

where $\mathbf{x}, \mathbf{y} \in \mathbb{D}_c^n, \mathbf{v} \in \mathcal{T}_{\mathbf{x}}\mathbb{D}_c^n$ and $\oplus_c$ is the Möbius addition. For convenience, we write $\mathbb{D}$ for $\mathbb{D}_c^n$ if no confusion arises.

A salient feature of hyperbolic geometry is that it is "thinner" than Euclidean geometry. Visually, more points can be squeezed in a hyperbolic subspace having the same shape as its Euclidean counterpart, due to the different metrics in the two spaces. We discuss the graph version in Section 3.1 below.

### 2.2 Graph attention and message passing

Consider a graph $G = (V, E)$, where $V$ is the set of vertices, $E$ is the set of edges, and each node in $V$ is associated with a node feature $h_v$. Recall that GAT is a GNN that updates node representations using

message passing by updating edge weights concurrently. Specifically, for one layer of GAT (Veličković et al., 2018), the node features are updated as follows:

$$h_v^{'} = \sigma\Big( \sum_{j \in N(v)} \alpha_{vj} \mathbf{W} h_j \Big), \tag{3}$$

$$\alpha_{vj} = \frac{\exp(e_{vj})}{\sum_{k \in N(v)} \exp(e_{vk})}, \tag{4}$$

$$e_{vj} = \text{LeakyReLU}(\mathbf{a}^{\intercal}[\mathbf{W} h_v \parallel \mathbf{W} h_j]), \tag{5}$$

where $\parallel$ denotes the concatenation operation, $\sigma$ denotes an activation function, $\mathbf{a}$ represents the learnable attention vector, $\mathbf{W}$ is the weight matrix for a linear transformation and $\alpha$ denotes the normalized attention scores.

This model has been proven to be successful in many graph-related machine learning tasks.

## 2.3 Hyperbolic attention model

To derive a hyperbolic version of GAT, we adopt the following strategy. We perform feature aggregation in the tangent spaces of points in the hyperbolic space. Features are mapped between hyperbolic space and tangent spaces using the pair of exponential and logarithmic functions: $\exp_{\mathbf{x}}^c$ and $\log_{\mathbf{x}}^c$.

With this, we denote Euclidean features as $h_{\mathbb{R}}$ and hyperbolic features as $h_{\mathbb{D}}$. Then one layer of message propagation in the hyperbolic GAT is as follows (Zhu et al., 2020):

$$h_{v,\mathbb{D}}^{'} = \sigma\Big( \sum_{j \in N(v)} \alpha_{vj} \log_{\mathbf{o}}^c(\mathbf{W} \otimes_c h_{j,\mathbb{D}} \oplus_c \mathbf{b}) \Big), \tag{6}$$

$$e_{vj} = \text{LeakyReLU}\Big( \mathbf{a}^{\intercal}\Big[ \hat{h}_v \parallel \hat{h}_j \Big] \times d_{\mathbb{D}}(h_{v,\mathbb{D}}, h_{j,\mathbb{D}}) \Big), \tag{7}$$

$$d_{\mathbb{D}}(h_{v,\mathbb{D}}, h_{j,\mathbb{D}}) = \frac{2}{\sqrt{c}} \tanh^{-1}(\sqrt{c}\| - h_{v,\mathbb{D}} \oplus_c h_{j,\mathbb{D}}\|), \tag{8}$$

$$\alpha_{vj} = \text{softmax}_j(e_{vj}), \tag{9}$$

where $d_{\mathbb{D}}$ is the normalized hyperbolic distance, $\hat{h}_j = \log_{\mathbf{o}}^c(\mathbf{W} \otimes_c h_{j,\mathbb{D}})$, while $\otimes_c$ and $\oplus_c$ represent the Möbius matrix multiplication and addition, respectively.

# 3 Joint Space Learning

In this section, we propose our joint space learning model. The model relies on comparing two different notions of hyperbolicity: geometric hyperbolicity and model hyperbolicity. We start by introducing the former, which also serves as the motivation for the design of our GNN model.

## 3.1 Local geometry and geometric hyperbolicity

Gromov's $\delta$-hyperbolicity is a mathematical notion from geometry group theory to measure how tree-like a metric space is in terms of metric or distance structure (Adcock et al., 2013; Chami et al., 2019). The precise definition is given as follows.

**Definition 1** (Gromov 4-point $\delta$-hyperbolicity (Bridson & Haefliger, 1999) p.410). *For a metric space $X$ with metric $d(\cdot, \cdot)$, it is $\delta$-hyperbolic, where $\delta \geq 0$ if the four-point condition holds:*

$$\begin{aligned} d(x,y) + d(z,t) &\leq \\ &\max\{d(x,z) + d(y,t), d(z,y) + d(x,t)\} + 2\delta, \end{aligned} \tag{10}$$

*for any $x, y, z, t \in X$. $X$ is hyperbolic if it is $\delta$-hyperbolic for some $\delta \geq 0$.*

This condition of $\delta$-hyperbolicity is equivalent to the Gromov thin triangle condition. For example, any tree is (0-)hyperbolic, and $\mathbb{R}^n$, where $n \geq 2$ is not hyperbolic. However, if $X$ is a compact metric space, then $X$ is always $\delta$-hyperbolic for some $\delta$ large enough such as $\delta = \text{diameter}(X)$. Therefore, it is insufficient to just label $X$ as hyperbolic or not. We want to quantify hyperbolicity such that a space with smaller hyperbolicity resembles more of a tree.

Inspired by the four-point condition, we define the $\infty$-version and the 1-version of hyperbolicity as follows.

**Definition 2.** *For a compact metric space $X$ and $x, y, z, t \in X$, denote $\inf_{\delta \geq 0}\{(10)$ holds for $x, y, z, t\}$ by $\tau_X(x, y, z, t)$. Define*

$$\delta_{X,\infty} = \sup_{x,y,z,t \in X} \tau_X(x, y, z, t),$$

$$\delta_{X,1} = \mathbb{E}_{x,y,z,t \sim \text{Unif}(X^4)}[\tau_X(x, y, z, t)],$$

*where* Unif *represents the uniform distribution.*

In order for these invariants to be useful for graphs, we require them to be almost identical for graphs with similar structures. We shall see that this is indeed the case. Before stating the result, we need a few more concepts.

Let $\mathcal{G}$ be the space of weighted, undirected simple graphs. Though for most experiments, the given graphs are unweighted. However, aggregation mechanisms such as attention essentially generate weights for the edges. Therefore, for both theoretical and practical reasons, it makes sense to expand the graph domain to include weighted graphs.

For each $G = (V, E) \in \mathcal{G}$, it has a canonical path metric $d_G$, and $d_G$ makes $G$ into a metric space including non-vertex points on the edges. For $\epsilon > 0$, there is the subspace $\mathcal{G}_\epsilon$ of $\mathcal{G}$ consisting of graphs whose edge weights are greater than $\epsilon$.

On the other hand, there is a metric on the space $\mathcal{G}$ and $\mathcal{G}_\epsilon$, called the Gromov-Hausdorff metric (Bridson & Haefliger (1999) p.72). To define it, we first introduce the Hausdorff distance. Let $X$ and $Y$ be two subsets of a metric space $(M, d)$. Then the Hausdorff distance $d_H(X, Y)$ between $X$ and $Y$ is

$$d_H(X, Y) = \max\{\sup_{x \in X} d(x, Y), \sup_{y \in Y} d(X, y)\},$$

where $d(x, Y) = \inf_{y \in Y} d(x, y)$, $d(X, y) = \inf_{x \in X} d(x, y)$. The Hausdorff distance measures in the worst case, how far away a point in $X$ is away from $Y$ and vice versa.

In general, we want to also compare spaces that do not a priori belong to a common ambient space. For this, if $X, Y$ are two compact metric spaces, then their Gromov-Hausdorff distance $d_{GH}(X, Y)$ is defined as the infimum of all numbers $d_H(f(X), g(Y))$ for all metric spaces $M$ and all isometric embeddings $f : X \to M, g : Y \to M$. Intuitively, the Gromov-Hausdorff distance measures how far $X$ and $Y$ are from being isometric. The following is proved in the Appendix.

**Proposition 1.** *Suppose $\mathcal{G}$ and its subspaces have the Gromov-Hausdorff metric. Then $\delta_{G,\infty}$ is Lipschitz continuous w.r.t. $G \in \mathcal{G}$ and $\delta_{G,1}$ is continuous w.r.t. $G \in \mathcal{G}_\epsilon$ for any $\epsilon > 0$.*

Consider a graph $G$. We fix either $\delta_{G,\infty}$ or $\delta_{G,1}$ as a measure of hyperbolicity, and apply to each local neighborhood of $G$. To be more precise, it is studied (Chen et al., 2020b; Rong et al., 2020) that many popular GNN models have a shallow structure. It is customary to have a 2-layer network possibly due to oversmoothing (Chen et al., 2020a; Chamberlain et al., 2021b; Zeng et al., 2021) and oversquashing (Topping et al., 2022) phenomena. In such models, each node only aggregates information in a small neighborhood.

Therefore, if we fix a small $k$ and let $G_v$ be the subgraph of the $k$-hop neighborhood of $v \in V$, then it is more appropriate to study the hyperbolicity $\delta_v$, either $\delta_{G_v,\infty}$ or $\delta_{G_v,1}$, of $G_v$. For our experiments, the former is utilized. We call $\delta_v$ the *geometric hyperbolicity* at node $v$. The collection $\Delta_V = \{\delta_v : v \in V\}$ allows us to obtain an empirical distribution $\mu_G$ of geometric hyperbolicity on the sample space $\mathbb{R}_{\geq 0}$.

For instance, we can build histograms to acquire the distributions as observed in Fig. 2. In Cora, we observe that a substantial number of nodes have small (local) hyperbolicity, despite its high global hyperbolicity value

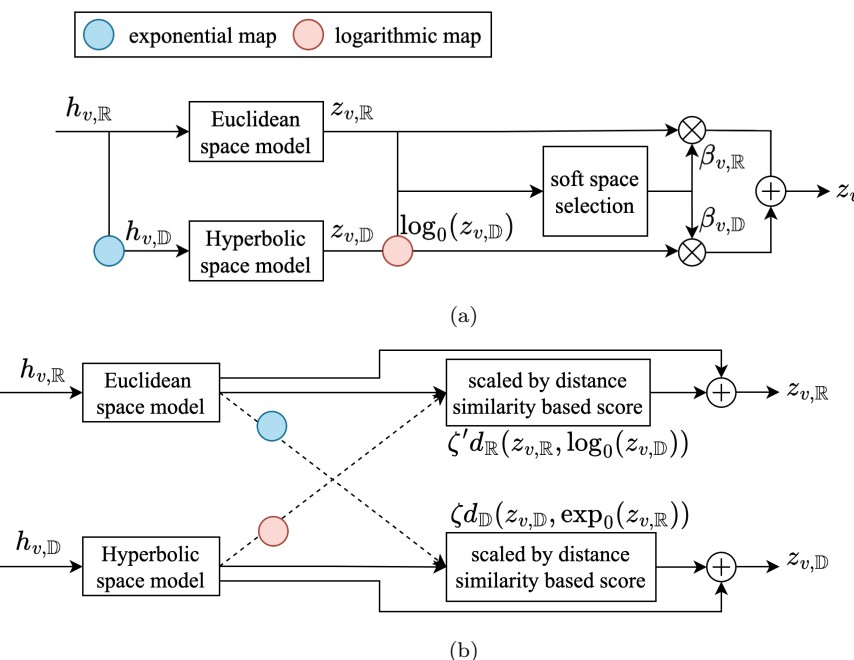

Figure 3: Comparison between JSGNN and GIL (Zhu et al., 2020) in leveraging Euclidean and hyperbolic spaces. Both models utilizes GAT in Section 2.2 and HGAT in Section 2.3 for message passing in Euclidean and hyperbolic space respectively. (a) Soft space selection mechanism of JSGNN where trainable selection weights $\beta_{v,\mathbb{R}}, \beta_{v,\mathbb{D}}$ are non-uniform, effectively selecting the better of the two spaces considered. (b) Feature interaction mechanism of GIL where $\zeta, \zeta' \in \mathbb{R}$ are trainable weights and $d_{\mathbb{D}}, d_{\mathbb{R}}$ are the hyperbolic distance (cf. (8)) and Euclidean distance respectively. The node embeddings of both spaces in GIL are adjusted based on distance, potentially introducing more noise to the branches as there is minimal information in the sub-optimal space to "enhance" the representation in the better space.

(Chami et al., 2019; Liu et al., 2022a). Meanwhile, Airport is argued to be globally hyperbolic, but a large proportion of nodes has large local hyperbolicity. However, this is not a contradiction as we are considering the local structures of the graph. We call $\mu_G$ the *distribution of geometric hyperbolicity*. It depends only on $G$ and $k$.

## 3.2 Space selection and model hyperbolicity

In this section, we describe the backbone of our model and introduce the notion of model hyperbolicity. Our model consists of two branches, one using Euclidean geometry and the other using hyperbolic geometry. We primarily use GAT (cf. Section 2.2) for the Euclidean model and HGAT (cf. Section 2.3) for the hyperbolic model. Other pairs of Euclidean or hyperbolic models (e.g., GCN and HGCN) can also be applied to the corresponding branches. In Section 4.3, we show the experimental results on two variants of JSGNN.

After the respective message propagation, we would have two sets of updated node embeddings, the Euclidean embedding $Z_{\mathbb{R}}$ and the hyperbolic embedding $Z_{\mathbb{D}}$. The two sets of embeddings are combined into a single embedding $Z = \{z_v, v \in V\}$ through an attention mechanism that serves as a space selection procedure. The attention mechanism is performed in a Euclidean space. Thus, the hyperbolic embeddings are first mapped into the tangent space using the logarithmic map. Mathematically, the normalized attention score indicating whether a node should be embedded in the hyperbolic space $\beta_{v,\mathbb{D}}$ or Euclidean space $\beta_{v,\mathbb{R}}$ is as follows:

$$w_{v,\mathbb{R}} = \mathbf{q}^{\mathsf{T}} \tanh(\mathbf{M} z_{v,\mathbb{R}} + \mathbf{b}), \tag{11}$$

$$w_{v,\mathbb{D}} = \mathbf{q}^{\mathsf{T}} \tanh(\mathbf{M} \log_{\mathbf{o}}^{c}(z_{v,\mathbb{D}}) + \mathbf{b}), \tag{12}$$

$$\beta_{v,\mathbb{R}} = \frac{\exp(w_{v,\mathbb{R}})}{\exp(w_{v,\mathbb{R}}) + \exp(w_{v,\mathbb{D}})}, \tag{13}$$

$$\beta_{v,\mathbb{D}} = \frac{\exp(w_{v,\mathbb{D}})}{\exp(w_{v,\mathbb{R}}) + \exp(w_{v,\mathbb{D}})}, \tag{14}$$

where $\mathbf{q}$ refers to the learnable space selection attention vector, $\mathbf{M}$ is a learnable weight matrix, $\mathbf{b}$ denotes a learnable bias and $\beta_{v,\mathbb{D}} + \beta_{v,\mathbb{R}} = 1$, for all $v \in V$. The weights $\beta_{v,\mathbb{D}}$ and $\beta_{v,\mathbb{R}}$ are conditioned to be non-uniform as illustrated in Section 3.4. The two sets of space-specific node embeddings can then be combined via a convex combination using the learned weights as follows:

$$z_v = \beta_{v,\mathbb{R}} z_{v,\mathbb{R}} + \beta_{v,\mathbb{D}} \log_{\mathbf{o}}^c(z_{v,\mathbb{D}}), \forall\, v \in V. \tag{15}$$

This gives one layer of the model architecture of JSGNN, as illustrated in Fig. 3.

The parameter $\beta_{v,\mathbb{R}}, v \in V$ controls whether the combined output, consisting of both hyperbolic and Euclidean components, should rely more on the hyperbolic components or not. We call $\beta_{v,\mathbb{R}}$ the *model hyperbolicity* at the node $v$. The notion of model hyperbolicity depends on node features as well as the explicit GNN model. Similar to geometric hyperbolicity, the collection $\Gamma_G = \{\beta_{v,\mathbb{R}} : v \in V\}$ gives rise to an empirical distribution $\nu_G$ on $[0, 1]$. We call $\nu_G$ the *distribution of model hyperbolicity.*

To motivate the next subsection, from (15), we notice that the output depends smoothly on $\beta_{v,\mathbb{R}}$. If we wish to have a similar output for nodes with similar neighborhood structures and features, we want their selection weights to have similar values. On the other hand, we have seen (cf. Proposition 1) that geometric hyperbolicities, which can be computed given $G$, are similar for nodes with similar neighborhoods. It suggests that we may use geometric hyperbolicities to "guide" the choice of model hyperbolicities.

### 3.3 Model hyperbolicity *vs.* geometric hyperbolicity

We have introduced geometric and model hyperbolicities in the previous subsections. In this subsection, we explore the interconnections between these two notions.

Let $\Theta$ be the parameters of a proposed GNN model. We assume that the model has the pipeline shown in Fig. 4. Given node features $\{h_v, v \in V\}$ and model parameters $\Theta$, the model generates (embedding) features $\{z_v, v \in V\}$ and selection weights or model hyperbolicity $\{\beta_{v,\mathbb{R}}, v \in V\}$ in the intermediate stage. For each $v \in V$, there is a combination function $\phi_v$ such that the final output $\{\hat{y}_v, v \in V\}$ satisfies $\hat{y}_v = \phi_v(z_v, \beta_v)$.

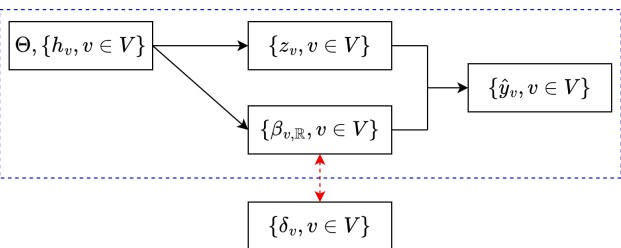

Figure 4: The model pipeline is shown in the (blue) dashed box, while the geometric hyperbolicity can be computed independently of the model.

In principle, we want to compare $\{\beta_{v,\mathbb{R}}, v \in V\}$ and $\{\delta_v, v \in V\}$ so that the geometric hyperbolicity guides the choice of model hyperbolicity. However, comparing pairwise $\beta_v$ and $\delta_v$ for each $v \in V$ may lead to overfitting. An alternative is to compare their respective distributions $\nu_G$ and $\mu_G$, or even coarser statistics (e.g., mean) of $\nu_G$ and $\mu_G$ (cf. Fig. 5). The latter may lead to underfitting. We perform an ablation study on the different comparison methods in Section 4.5.

We advocate choosing the middle ground by comparing the distributions $\mu_G$ and $\nu_G$. The former can be computed readily as long as the ambient graph $G$ is given, while the latter is a part of the model that plays a crucial role in feature aggregation at each node. Therefore, $\mu_G$ can be pre-determined but not $\nu_G$. We propose to use the known $\mu_G$ to constrain $\nu_G$ and thus the model parameters $\Theta$. A widely used comparison tool is the Wasserstein metric.

$$\{\beta_{v,\mathbb{R}}, v \in V\} \;\mapsto\; \nu_G \;\mapsto\; \text{Stat. of } \nu_G$$

$$\{\delta_v, v \in V\} \;\mapsto\; \mu_G \;\mapsto\; \text{Stat. of } \mu_G$$

Figure 5: Different ways of comparing geometric and model hyperbolicities.

**Definition 3** (Wasserstein distance). *Given $p \geq 1$, the p-Wasserstein distance metric Villani (2008) measures the difference between two different probability distributions Gao et al. (2021). Let $\Pi(\nu_G, \mu_G)$ be the set of all joint distributions for random variables $x$ and $y$ where $x \sim \nu_G$ and $y \sim \mu_G$. Then the p-Wasserstein distance between $\mu_G$ and $\nu_G$ is as follows:*

$$W_p(\nu_G, \mu_G) = \left\{ \inf_{\gamma \in \Pi(\nu_G, \mu_G)} \mathbb{E}_{(x,y) \sim \gamma} \|x - y\|^p \right\}^{1/p}. \tag{16}$$

To compute the Wasserstein distance exactly is costly given that the solution of an optimal transport problem is required (Rowland et al., 2019; Chen et al., 2022b). However, for one-dimensional distributions, the *p*-Wasserstein distance can be computed by ordering the *samples* from the two distributions and then computing the average *p*-distance between the ordered samples (Kolouri et al., 2019; Rowland et al., 2019).

In ideal circumstances, considering the distributions do not lose much information. We first notice that for both $\beta_{v,\mathbb{R}}$ and $\delta_v$, a smaller value means more hyperbolic in an appropriate sense. Suppose $\beta_{v,\mathbb{R}}$ is increasing w.r.t. $\delta_v$, i.e., $\delta_v \leq \delta_u$ implies that $\beta_{v,\mathbb{R}} \leq \beta_{u,\mathbb{R}}$. Then, $W_2(\mu_G, \nu_G) = \sqrt{\frac{1}{|V|} \sum_{v \in V} |\beta_{v,\mathbb{R}} - \delta_v|^2}$.

### 3.4 Non-uniformity of selection weights

A node is considered to be more suitable to be embedded in the hyperbolic space when $\beta_{v,\mathbb{D}} > \beta_{v,\mathbb{R}}$. Meanwhile when $\beta_{v,\mathbb{D}} \leq \beta_{v,\mathbb{R}}$, the node is considered to be Euclidean. Nevertheless, to align with our motivation that each node can be better embedded in one of the two spaces and the less suitable space would result in distortion in representation, we require JSGNN to learn non-uniform attention weights, meaning that each pair of attention weights $(\beta_{v,\mathbb{D}}, \beta_{v,\mathbb{R}})$ should significantly deviate from the uniform distribution. This is because soft selection without a non-uniformity constraint may result in the assignment of nodes to be partially Euclidean and partially hyperbolic with $\beta_{v,\mathbb{R}} \approx \beta_{v,\mathbb{D}} \approx 0.5$. Hence, we include an additional component to the standard loss function encouraging non-uniform learned weights as follows:

$$L_{\text{nu}} = -\frac{1}{|V|} \sum_{v \in V} \left( \beta_{v,\mathbb{R}}^2 + \beta_{v,\mathbb{D}}^2 \right). \tag{17}$$

Since $-1 \leq -(\beta_{v,\mathbb{R}}^2 + \beta_{v,\mathbb{D}}^2) \leq -0.5$ and $\beta_{v,\mathbb{R}} + \beta_{v,\mathbb{D}} = 1$, minimizing the term would favor non-uniform attention weights for each node.

In summary, we may combine hyperbolicity matching discussed in Section 3.3 and the non-uniformity loss to form the loss function to optimize JSGNN.

$$L_{\text{overall}} = L_{\text{task}} + \omega_{\text{nu}} L_{\text{nu}} + \omega_{\text{was}} W_2(\nu_G, \mu_G), \tag{18}$$

where $L_{\text{task}}$ is the task-specific loss, while $\omega_{\text{nu}}$ and $\omega_{\text{was}}$ are balancing factors. For the node classification task, $L_{\text{task}}$ refers to the cross-entropy loss over all labeled nodes while for link prediction, it refers to the cross-entropy loss with negative sampling. This completes the description of the JSGNN model.

We speculate that the non-uniform component $L_{\text{nu}}$ should push the model hyperbolicities towards the two extremes 0 and 1. On the other hand, as we have seen in Section 3.3, to compute $W_2(\nu_G, \mu_G)$, we need to order $(\delta_v)_{v \in V}$, $(\beta_{v,\mathbb{R}})_{v \in V}$ respectively, and compute their pairwise differences. Therefore, $W_2(\nu_G, \mu_G)$ aligns the shapes of $\nu_G$ and $\mu_G$.

Table 1: Node classification result on Cora, Citeseer and Pubmed datasets. Performance score averaged over ten runs. The best performance is boldfaced while the second-best performance is underlined.

| Method | Standard split | | | 60/20/20% split | | |
|---|---|---|---|---|---|---|
| | Cora | Citeseer | Pubmed | Cora | Citeseer | Pubmed |
| GCN | $81.53 \pm 0.84$ | $70.47 \pm 0.64$ | $78.30 \pm 0.63$ | $91.99 \pm 0.79$ | $84.13 \pm 0.98$ | $88.79 \pm 1.63$ |
| GAT | $81.68 \pm 1.06$ | $70.96 \pm 0.96$ | $78.05 \pm 0.50$ | $91.63 \pm 0.57$ | $83.93 \pm 0.85$ | $\underline{89.99 \pm 1.31}$ |
| GraphSAGE | $76.59 \pm 1.06$ | $65.26 \pm 2.91$ | $77.90 \pm 0.71$ | $91.25 \pm 0.22$ | $84.08 \pm 0.25$ | $89.62 \pm 0.18$ |
| CurvGN | $81.58 \pm 0.51$ | $71.14 \pm 0.67$ | $78.17 \pm 0.48$ | $91.60 \pm 0.25$ | $84.18 \pm 0.37$ | $86.65 \pm 0.14$ |
| CGNN | $\underline{82.15 \pm 0.60}$ | $\underline{71.31 \pm 1.16}$ | $78.34 \pm 0.83$ | $91.96 \pm 0.27$ | $84.29 \pm 0.34$ | $86.86 \pm 0.16$ |
| HGNN | $79.28 \pm 0.77$ | $70.00 \pm 0.74$ | $77.45 \pm 1.40$ | $89.73 \pm 0.84$ | $80.27 \pm 0.21$ | $88.27 \pm 0.51$ |
| HGCN | $78.68 \pm 0.77$ | $67.25 \pm 1.45$ | $76.72 \pm 0.92$ | $91.57 \pm 0.28$ | $83.68 \pm 0.52$ | $86.83 \pm 0.31$ |
| HGAT | $78.81 \pm 1.49$ | $68.16 \pm 1.34$ | $77.43 \pm 1.20$ | $90.27 \pm 0.81$ | $81.29 \pm 0.79$ | $86.27 \pm 0.47$ |
| LGCN | $78.93 \pm 0.79$ | $68.59 \pm 0.64$ | $78.08 \pm 0.65$ | $92.55 \pm 0.57$ | $\underline{85.03 \pm 0.28}$ | $89.59 \pm 0.11$ |
| $\kappa$-GCN ($\mathbb{D}^{16}$) | $78.64 \pm 0.83$ | $67.17 \pm 0.73$ | $78.01 \pm 0.67$ | $89.16 \pm 0.59$ | $84.57 \pm 0.20$ | $88.37 \pm 0.37$ |
| DeepHGCN | $80.66 \pm 0.89$ | $\mathbf{72.11 \pm 0.60}$ | $78.13 \pm 1.67$ | $88.51 \pm 1.52$ | $80.45 \pm 0.36$ | $86.90 \pm 0.42$ |
| $\kappa$-GCN ($\mathbb{D}^{16} \times \mathbb{R}^{16}$) | $78.71 \pm 1.02$ | $66.96 \pm 1.13$ | $77.67 \pm 0.74$ | $88.85 \pm 0.88$ | $84.04 \pm 0.81$ | $85.59 \pm 0.53$ |
| GIL | $79.97 \pm 1.93$ | $67.54 \pm 1.23$ | $76.62 \pm 0.81$ | $91.90 \pm 0.84$ | $82.39 \pm 0.90$ | $87.39 \pm 0.21$ |
| JSGNN (GCN+HGCN) | $81.79 \pm 0.80$ | $70.55 \pm 1.09$ | $\underline{78.38 \pm 0.74}$ | $\mathbf{93.26 \pm 0.92}$ | $84.95 \pm 0.31$ | $89.68 \pm 0.60$ |
| JSGNN (GAT+HGAT) | $\mathbf{82.94 \pm 0.55}$ | $71.26 \pm 1.13$ | $\mathbf{78.57 \pm 0.90}$ | $\underline{93.10 \pm 0.86}$ | $\mathbf{85.10 \pm 0.64}$ | $\mathbf{90.53 \pm 0.32}$ |

## 4 Experiments

In this section, we evaluate JSGNN on node classification (NC) and link prediction (LP) tasks. Dataset statistics, model settings, and model size and complexity are discussed in Appendix A and Appendix B.

### 4.1 Datasets

A total of seven benchmark datasets are employed for both NC and LP. Specifically, three citation datasets: Cora, Citeseer, Pubmed; a flight network: Airport; a disease propagation tree: Disease; an Amazon co-purchase graph dataset: Photo; and a coauthor dataset: CS.

### 4.2 Baselines

For JSGNN, we consider two variants: GAT as the Euclidean model with HGAT as the hyperbolic model, and GCN as the Euclidean model with HGCN as the hyperbolic model. We compare against the following models: (a) Euclidean methods: GCN (Kipf & Welling, 2016), GraphSAGE (Hamilton et al., 2017) and GAT (Veličković et al., 2018); (b) hyperbolic models: HGCN (Chami et al., 2019), HGNN (Liu et al., 2019), HGAT (Zhang et al., 2021a), LGCN (Zhang et al., 2021b), DeepHGCN (Liu et al., 2024) and $\kappa$-GCN ($\mathbb{D}^{16}$) (Bachmann et al., 2019), which is a constant (negative) curvature graph neural network based on the $\kappa$-stereographic model; (c) CurvGN (Ye et al., 2020) and CGNN (Li et al., 2021), which utilizes graph curvature information to filter messages differently based upon different local structures; (d) mixed models: GIL (Zhu et al., 2020) and $\kappa$-GCN ($\mathbb{D}^{16} \times \mathbb{R}^{16}$), which similar to JSGNN, leverages multiple spaces or a mixed curvature space. $\kappa$-GCN ($\mathbb{D}^{16} \times \mathbb{R}^{16}$) employs a two-component product space of negative and zero curvature.

### 4.3 Node classification

For the node classification task, each of the nodes in a dataset belongs to one of the $C$ classes in the dataset. With the final set of node representations, we aim to predict the labels of nodes that are in the testing set.

To test the performance of each model under both semi-supervised and fully-supervised settings, two data splits are used in the node classification task for the Cora, Citeseer and Pubmed datasets. In the first split, we followed the standard split for semi-supervised settings used in Kipf & Welling (2016); Veličković et al. (2018); Monti et al. (2017); Chamberlain et al. (2021b); Zhu et al. (2020); Chamberlain et al. (2021a); Hamilton et al. (2017); Liu et al. (2022b); Feng et al. (2020). The train set consists of 20 train examples per class while the

Table 2: Node classification result on CS, Photo, Airport and Disease datasets. OOM corresponds to out-of-memory.

| Method | CS | Photo | Airport | Disease |
|---|---|---|---|---|
| GCN | $96.53 \pm 0.10$ | $94.05 \pm 0.27$ | $79.62 \pm 1.28$ | $83.32 \pm 1.37$ |
| GAT | $96.36 \pm 0.38$ | $94.45 \pm 0.92$ | $83.07 \pm 1.52$ | $86.05 \pm 1.08$ |
| GraphSAGE | $96.45 \pm 0.91$ | $96.13 \pm 1.61$ | $81.73 \pm 0.98$ | $83.47 \pm 1.77$ |
| CurvGN | $96.91 \pm 0.09$ | $94.21 \pm 0.22$ | $87.25 \pm 0.88$ | $84.35 \pm 3.20$ |
| CGNN | $96.27 \pm 0.08$ | $93.93 \pm 0.18$ | $87.21 \pm 1.08$ | $85.75 \pm 2.08$ |
| HGNN | $96.72 \pm 0.19$ | $94.74 \pm 0.66$ | $84.37 \pm 1.19$ | $87.40 \pm 1.66$ |
| HGCN | $96.58 \pm 0.10$ | $95.27 \pm 0.25$ | $89.39 \pm 1.52$ | $87.93 \pm 1.61$ |
| HGAT | $96.65 \pm 0.15$ | $96.62 \pm 0.28$ | $89.31 \pm 1.09$ | $90.04 \pm 1.50$ |
| LGCN | OOM | $\underline{96.71 \pm 0.24}$ | $88.53 \pm 1.26$ | $\underline{91.15 \pm 1.02}$ |
| $\kappa$-GCN ($\mathbb{D}^{16}$) | $97.04 \pm 0.10$ | $95.56 \pm 0.54$ | $89.08 \pm 1.15$ | $\mathbf{92.39 \pm 0.73}$ |
| DeepHGCN | $95.80 \pm 0.88$ | $94.37 \pm 1.43$ | $90.24 \pm 1.88$ | $90.38 \pm 1.76$ |
| $\kappa$-GCN ($\mathbb{D}^{16} \times \mathbb{R}^{16}$) | $96.97 \pm 0.10$ | $94.31 \pm 0.41$ | $88.38 \pm 0.62$ | $89.47 \pm 1.56$ |
| GIL | $95.83 \pm 0.30$ | $94.41 \pm 0.57$ | $\mathbf{90.78 \pm 1.74}$ | $90.67 \pm 1.98$ |
| JSGNN (GCN+HGCN) | $\underline{97.08 \pm 0.04}$ | $95.69 \pm 0.22$ | $\underline{90.59 \pm 1.75}$ | $90.73 \pm 1.44$ |
| JSGNN (GAT+HGAT) | $\mathbf{97.40 \pm 0.14}$ | $\mathbf{97.16 \pm 0.44}$ | $90.33 \pm 1.61$ | $90.88 \pm 1.54$ |

validation set and test set consist of 500 samples and 1,000 samples, respectively.[1] Meanwhile, in the second split, all labels are utilized and the percentages of training, validation, and test sets are set as 60/20/20%. For the Photo and CS datasets, the labeled nodes are also split into three sets where 60% of the nodes made up the training set, and the rest of the nodes were divided equally to form the validation and test sets. Airport and Disease datasets were split in similar settings as Zhu et al. (2020).

In Table 1 and Table 2, the mean accuracy with standard deviation is reported for node classification, except for the case of Airport and Disease datasets where the mean F1 score is reported. Our empirical results demonstrate that JSGNN frequently outperforms the baselines, especially HGAT and GAT which are the building blocks of JSGNN. Even though the performance of the variant JSGNN (GCN+HGCN) is often slightly lower than JSGNN (GAT+HGAT), we have similarly observed it to consistently outperform its building blocks GCN and HGCN. This is not necessarily observed for other mixed space models. Thus, this shows the superiority of not only using both Euclidean and hyperbolic spaces but also our method of incorporating the two spaces for graph learning as compared to GIL and $\kappa$-GCN ($\mathbb{D}^{16} \times \mathbb{R}^{16}$).

We also observe that Euclidean models such as GCN, GAT, GraphSAGE, CurvGN and CGNN perform better than hyperbolic models in general on the Cora, Citeseer, and Pubmed datasets for both splits. Meanwhile, hyperbolic models achieve better results on the CS, Photo, Airport, and Disease datasets. This means that Euclidean features are more significant for representing Cora, Citeseer and Pubmed datasets while hyperbolic features are more significant for the others. Nevertheless, JSGNN is able to perform relatively well across all datasets. We note that JSGNN exceeds the performance of single-space baselines on all datasets except for Disease. This can be explained by the fact that Disease consists of a perfect tree and thus, does not exhibit different hyperbolicities in the graph. However, JSGNN still outperforms 3 hyperbolic benchmarks and all the other mixed models.

We also particularly note that the difference in results between single-space models using only the Euclidean embedding space and hyperbolic models is not significant. This means that many of the node labels can be potentially predicted even without the best representation from the right space. This might be the reason why the gain in performance for the node classification task is not exceptional from embedding nodes in the better space. Nevertheless, we still see improvements in predictions for cases where there is a mixture of local hyperbolicities. Moreover, embedding nodes in a more suitable space can benefit other tasks that require more accurate representations such as link prediction.

---

[1]Note that the top results on `https://paperswithcode.com/sota/node-classification-on-cora` used different data splits (either semi-supervised settings with a larger number of training samples or fully-supervised settings such as the 60/20/20% split) which give much higher accuracies

Table 3: Link prediction result averaged over ten runs.

| Method | Cora | Citeseer | Pubmed | Airport | Disease |
|---|---|---|---|---|---|
| GCN | $88.22 \pm 1.01$ | $90.60 \pm 1.10$ | $87.63 \pm 3.25$ | $91.79 \pm 1.48$ | $61.60 \pm 3.76$ |
| GAT | $85.47 \pm 2.28$ | $85.31 \pm 1.89$ | $85.30 \pm 1.46$ | $93.70 \pm 0.65$ | $61.23 \pm 2.75$ |
| GraphSAGE | $88.94 \pm 0.81$ | $91.61 \pm 1.00$ | $88.42 \pm 1.14$ | $91.63 \pm 0.81$ | $68.31 \pm 2.94$ |
| CurvGN | $94.40 \pm 2.13$ | $95.38 \pm 2.33$ | $94.55 \pm 0.89$ | $93.90 \pm 0.37$ | $95.47 \pm 0.81$ |
| CGNN | $94.26 \pm 1.36$ | $96.54 \pm 0.78$ | $94.71 \pm 3.14$ | $95.13 \pm 0.97$ | $95.35 \pm 1.22$ |
| HGNN | $91.48 \pm 0.38$ | $93.63 \pm 0.14$ | $92.95 \pm 0.35$ | $96.31 \pm 0.30$ | $82.98 \pm 0.98$ |
| HGCN | $93.72 \pm 0.26$ | $96.72 \pm 1.69$ | $96.68 \pm 0.04$ | $97.55 \pm 0.08$ | $87.14 \pm 1.34$ |
| HGAT | $94.06 \pm 0.11$ | $95.60 \pm 0.20$ | $95.78 \pm 0.05$ | $97.86 \pm 0.08$ | $86.61 \pm 1.67$ |
| LGCN | $93.10 \pm 0.30$ | $93.40 \pm 0.70$ | $95.45 \pm 0.08$ | $97.88 \pm 0.19$ | $95.99 \pm 0.58$ |
| $\kappa$-GCN ($\mathbb{D}^{16}$) | $92.43 \pm 0.63$ | $94.38 \pm 0.51$ | $94.89 \pm 0.07$ | $96.78 \pm 0.19$ | $93.58 \pm 0.31$ |
| $\kappa$-GCN ($\mathbb{D}^{16} \times \mathbb{R}^{16}$) | $91.32 \pm 0.38$ | $92.87 \pm 0.27$ | $93.53 \pm 0.06$ | $97.17 \pm 0.10$ | $90.15 \pm 0.77$ |
| GIL | $98.04 \pm 1.64$ | $99.95 \pm 0.09$ | $92.50 \pm 0.50$ | $97.20 \pm 1.04$ | $\mathbf{100.00 \pm 0.00}$ |
| JSGNN (GCN+HGCN) | $\underline{98.83 \pm 0.92}$ | $\underline{99.97 \pm 0.09}$ | $\mathbf{97.67 \pm 0.03}$ | $\underline{98.94 \pm 0.91}$ | $\mathbf{100.00 \pm 0.00}$ |
| JSGNN (GAT+HGAT) | $\mathbf{99.43 \pm 0.21}$ | $\mathbf{99.98 \pm 0.05}$ | $\underline{96.95 \pm 0.03}$ | $\mathbf{99.26 \pm 1.23}$ | $\underline{99.97 \pm 0.08}$ |

Table 4: Ablation study of JSGNN (GAT+HGAT) for node classification task. Cora, Citeseer and Pubmed on the standard split.

| Method | CS | Photo | Cora | Citeseer | Pubmed | Airport | Disease |
|---|---|---|---|---|---|---|---|
| JSGNN (GAT+HGAT) | $\mathbf{97.40 \pm 0.14}$ | $\mathbf{97.16 \pm 0.44}$ | $\mathbf{82.94 \pm 0.55}$ | $\mathbf{71.26 \pm 1.13}$ | $\mathbf{78.57 \pm 0.90}$ | $\mathbf{90.33 \pm 1.61}$ | $\mathbf{90.88 \pm 1.54}$ |
| w/o NU & $W_2$ | $97.15 \pm 0.10$ | $95.95 \pm 0.41$ | $81.65 \pm 1.08$ | $70.87 \pm 1.22$ | $78.14 \pm 1.02$ | $89.67 \pm 1.26$ | $89.85 \pm 1.61$ |
| w/o $W_2$ | $97.33 \pm 0.20$ | $96.51 \pm 0.67$ | $82.36 \pm 0.78$ | $71.15 \pm 1.17$ | $78.50 \pm 0.53$ | $90.02 \pm 1.63$ | $90.66 \pm 2.22$ |
| w/o NU | $97.38 \pm 0.15$ | $96.42 \pm 0.37$ | $82.67 \pm 0.51$ | $70.86 \pm 1.45$ | $78.48 \pm 0.47$ | $89.98 \pm 1.72$ | $90.37 \pm 2.12$ |

## 4.4 Link prediction

We employ the Fermi-Dirac decoder with a distance function to model the probability of an edge based on our final output embedding, similar to Zhu et al. (2020); Sun et al. (2022); Chami et al. (2019). The probability that an edge exists is given by $\mathbb{P}(e_{vj} \in E \,|\, \Theta) = (e^{(d(x_i, x_j) - r)/t} + 1)^{-1}$ where $r, t > 0$ are hyperparameters and $d$ is the distance function. The edges of the datasets are randomly split into $85/5/10\%$ for training, validation, and testing. The average ROC AUC for link prediction is recorded in Table 3. We observe that JSGNN (GAT+HGAT) performs better than the baselines in most cases. For the link prediction task, we notice that hyperbolic models consistently outperform Euclidean models by a significant margin. Moreover, Euclidean methods such as CurvGN and CGNN benefit from using topological information during learning. Empirical results also suggest that predicting the existence of edges seems to benefit from dual space models, i.e., GIL and JSGNN, except for the case of $\kappa$-GCN ($\mathbb{D}^{16} \times \mathbb{R}^{16}$).

This finding is similar to that reported in Bachmann et al. (2019); Xiong et al. (2022) where despite slightly different settings, the *constant (negative) curvature $\kappa$-stereographic model frequently outperforms the $\kappa$-GCN leveraging on the product of multiple constant curvature spaces*. We hypothesize that the simple concatenation to combine the embeddings of the two different component spaces in $\kappa$-GCN ($\mathbb{D}^{16} \times \mathbb{R}^{16}$) might be insufficient and might have resulted in noise from the other space being passed to the negatively curved space which was performing well standalone as seen in $\kappa$-GCN ($\mathbb{D}^{16}$). As such, our aim to learn to select the better space for each node is potentially capable of offering better representations with reduced distortions.

## 4.5 Ablation study

We conduct an ablation study on the node classification task by introducing three variants of JSGNN (GAT+HGAT) to validate the effectiveness of the different components introduced:

- Without the non-uniformity constraint (w/o NU): This does not enforce the model to learn non-uniform selection weights.

- Without the Wasserstein metric (w/o $W_2$): The learning of model hyperbolicity is not guided by geometric hyperbolicity.

- Without the non-uniformity loss and Wasserstein distance (w/o NU & $W_2$): Only guided by the cross entropy loss, i.e., $\omega_{\mathrm{nu}} = 0, \omega_{\mathrm{was}} = 0$ (cf. (18)).

Table 4 summarizes the results of our study, from which we observe that all variants of JSGNN (GAT+HGAT) with some components discarded perform worse than the full model. Moreover, JSGNN (GAT+HGAT) without $W_2$ always achieves better results than JSGNN (GAT+HGAT) without NU and $W_2$, signifying the importance of selecting the better of the two spaces instead of combining the features with relatively uniform weights. Similarly, JSGNN (GAT+HGAT) without NU performs better than JSGNN (GAT+HGAT) without NU and $W_2$ in most cases, suggesting that incorporating geometric hyperbolicity through distribution alignment does help to improve the model. A comparison of the time taken for each variant is provided in Appendix B.

To further analyze our model, we present a study regarding our method of incorporating the guidance of geometric hyperbolicity through distribution alignment. The result is as seen in Table 5. We test and analyze empirically different variants of our model based on the different comparisons shown in Fig. 5. Pairwise match indicates minimizing the mean squared error between elements of $\Gamma_G$ and $\Delta_V$ (without sorting) while mean match minimizes the squared loss between the means of $\Gamma_G$ and $\Delta_V$. We observe that comparing the distributions of $\nu_G$ and $\mu_G$ consistently outperforms comparing their mean, demonstrating the insufficiency of utilising coarse statistics for supervision. Secondly, pairwise matching gave better results than mean matching, though still lower than distribution matching, suggesting the importance of fine-scale information yet, a need to avoid potential overfitting.

Table 5: Node classification results of different comparison methods to incorporate geometric hyperbolicity to guide model hyperbolicity.

| Dataset | Pairwise match | Distribution | Mean match |
|---|---|---|---|
| Cora | $82.35 \pm 1.06$ | $\mathbf{82.94 \pm 0.55}$ | $81.36 \pm 1.50$ |
| Citeseer | $70.06 \pm 2.05$ | $\mathbf{71.26 \pm 1.13}$ | $69.64 \pm 1.18$ |
| Pubmed | $78.46 \pm 0.86$ | $\mathbf{78.57 \pm 0.90}$ | $78.08 \pm 0.62$ |
| Aiport | $90.13 \pm 1.53$ | $\mathbf{90.33 \pm 1.61}$ | $89.31 \pm 2.22$ |
| Disease | $90.66 \pm 1.91$ | $\mathbf{90.88 \pm 1.54}$ | $87.53 \pm 6.24$ |
| Photo | $96.17 \pm 0.23$ | $\mathbf{97.16 \pm 0.44}$ | $95.96 \pm 0.59$ |
| CS | $97.20 \pm 0.16$ | $\mathbf{97.40 \pm 0.14}$ | $97.17 \pm 0.11$ |

### 4.6 Analysis of hyperbolicities

We have speculated the effects of different components of our proposed model at the end of Section 3.4. To verify that our model can learn model hyperbolicity that is non-uniform and similar in distribution as geometric hyperbolicity, we analyze the learned model hyperbolicities $(\beta_{v,\mathbb{R}})_{v \in V}$ of JSGNN (GAT+HGAT) and the model w/o NU & $W_2$ for the node classification task. Specifically, we extract the learned values from the first two layers of JSGNN and its variant for ten separate runs. The learned values from the first two layers were then averaged before determining $W_2(\nu_G, \mathrm{Unif})$ and $W_2(\nu_G, \mu_G)$.

In Fig. 6, it can be inferred that JSGNN's learned model hyperbolicity is always less uniform than that of the model w/o NU & $W_2$ given JSGNN's larger $W_2(\nu_G, \mathrm{Unif})$ score, demonstrating a divergence from uniform distribution. Meanwhile, for most cases, JSGNN's $W_2(\nu_G, \mu_G)$ is smaller than that of the model w/o NU & $W_2$, suggesting that the shape between $\nu_G$ and $\mu_G$ of JSGNN is relatively more similar. At times, JSGNN's $W_2(\nu_G, \mu_G)$ is larger than the model w/o NU & $W_2$, suggesting a tradeoff between NU and $W_2$ as we choose the optimal combination for the model's best performance.

## 5 Conclusion

In this paper, we have explored the learning of GNNs in a joint space setting given that different regions of a graph can have different geometrical characteristics. In these situations, it would be beneficial to embed different regions of the graph in different spaces that are better suited for their underlying structures, to

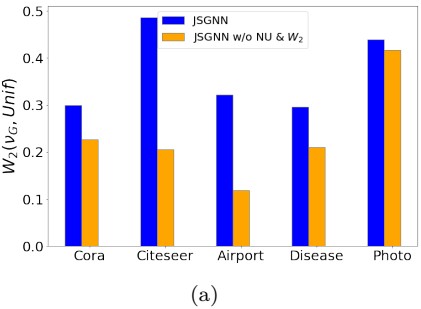 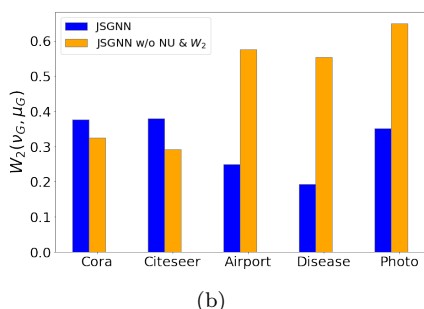

Figure 6: Analysis of hyperbolicities on different datasets. (a) $W_2(\nu_G, \text{Unif})$. (b) $W_2(\nu_G, \mu_G)$.

reduce the distortions incurred while learning node representations. Our method JSGNN utilizes a soft attention mechanism with non-uniformity constraint and distribution alignment between model and geometric hyperbolicities to select the best space-specific feature for each node. This indirectly finds the space that is best suited for each node. Experimental results of node classification and link prediction demonstrate the effectiveness of JSGNN against various baselines. In future work, we aim to further improve our model with an adaptive mechanism to determine the appropriate, node-level specific neighborhood to account for each node's hyperbolicity. Limitations are discussed in Appendix F.

## 6 Acknowledgements

The first author is supported by Shopee Singapore Private Limited under the Economic Development Board Industrial Postgraduate Programme (EDB IPP). The programme is a collaboration between Shopee and Nanyang Technological University, Singapore. This research is supported in part by the Singapore Ministry of Education Academic Research Fund Tier 2 grant MOE-T2EP20220-0002. This work is carried out during the first author's participation in the EDB IPP.

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

## A Dataset statistics and model settings

Dataset statistics are provided in Table 6.

Table 6: Dataset statistics.

| Dataset | Nodes | Edges | Classes | Features |
|---|---|---|---|---|
| Cora | 2708 | 5429 | 7 | 1433 |
| Citeseer | 3327 | 4732 | 6 | 3703 |
| Pubmed | 19717 | 44338 | 3 | 500 |
| Aiport | 3188 | 18631 | 4 | 4 |
| Disease | 1044 | 1043 | 2 | 1000 |
| Photo | 7650 | 119081 | 8 | 745 |
| CS | 18333 | 81894 | 15 | 6805 |

For all models, the hidden units are set to 16. We set the early stopping patience to 100 epochs with a maximum limit of 2000 epochs. The hyperparameter settings for the baselines are the same as Zhu et al. (2020) if given. The only difference is that the hyperparameter *h-drop* for GIL in Zhu et al. (2020) (which determines the dropout to the weight associated with the hyperbolic space embedding) is set to 0 for all datasets as setting a large value essentially explicitly chooses one single space. Else, the hyperparameters are chosen to yield the best performance. For JSGNN, we perform a grid search on the following search spaces: Learning rate: [0.01, 0.005]; Dropout probability: [0.0, 0.1, 0.5, 0.6]; Number of layers: [1, 2, 3]; $\omega_{\mathrm{nu}}$ and $\omega_{\mathrm{was}}$: [1.0, 0.5, 0.2, 0.1, 0.01, 0.005]; $\mathbf{q}$ (cf. (11)): [16, 32, 64]. The Wasserstein-2 distance is employed in all variants of JSGNN.

The source code can be found at `https://github.com/amblee0306/JSGNN_Mixed_Space_GNN`. The implementation is based on the setup in `https://github.com/CheriseZhu/GIL`.

## B Complexity, run-time and model size

We first incur an overhead to compute the geometric hyperbolicity $\delta_v$ for each node. We follow the approach as described below[2] rather than using the naive implementation with a time complexity of $O(V^4)$. Instead of considering all possible 4-tuples in the graph, we heuristically sample $K$ 4-tuples from each nodes' two-hop subgraph. Consequently, the complexity is $O(N) \times [O(N^2) + O(K \times E_{subgraph})] \approx O(NKd^2)$ where $d$ denotes the average degree in the graph, $K$ denotes the number of samples and $E_{subgraph}$ represents the number of edges in the two-hop subgraph of each node. Assuming the graph is sparse, $d$ is small. This runtime can be further optimized by parallelization, as each node's calculation is independent. Also, notice that this step is performed only once during pre-processing, and the step is model-agnostic. For a dataset, the computed $\{\delta_v\}_{v \in V}$ can be stored and re-used for different trainings and also for different base models. Moreover, if the dataset is updated, we only need to adjust $\delta_v$ for nodes $v$ whose neighborhood structures are changed during the update, which can be done efficiently.

For training of the model, we usually use two base models $\mathcal{M}_e$ (e.g., GCN) and $\mathcal{M}_h$ (e.g., HGCN) for handling Euclidean and hyperbolic structures respectively. The complexity of the message passing in our model is $O(C_e + C_h)$, where $C_e$ and $C_h$ are the message passing complexities of $\mathcal{M}_e$ and $\mathcal{M}_h$ respectively.

Regarding the implementation of the extra loss terms, calculating the Wasserstein distance for one-dimensional distributions requires a time complexity of $O(N \log N)$ where $N$ is the number of nodes in the graph, primarily dominated by the time needed to sort the distributions. Meanwhile, obtaining the non-uniformity loss has a time complexity of $O(N)$.

In Table 7, we present the time taken for 300 epochs across different variants of JSGNN that we examine in our ablation study in Section 4.5. The results show that, despite the added time complexity introduced by

---

[2]https://github.com/HazyResearch/hgcn/blob/master/utils/hyperbolicity.py

Table 7: Time taken (in seconds) for 300 epochs for node classification task across different model variants.

| Method | Cora | Citeseer | Pubmed | CS | Photo |
|---|---|---|---|---|---|
| JSGNN (GAT+HGAT) | $18.53 \pm 1.05$ | $21.60 \pm 0.62$ | $24.73 \pm 3.49$ | $49.35 \pm 0.62$ | $37.03 \pm 0.51$ |
| w/o $W_2$ | $17.29 \pm 0.66$ | $20.06 \pm 0.27$ | $23.56 \pm 1.85$ | $47.31 \pm 0.34$ | $34.45 \pm 0.41$ |
| w/o NU | $18.06 \pm 0.13$ | $19.98 \pm 0.32$ | $23.42 \pm 3.49$ | $48.36 \pm 0.41$ | $36.71 \pm 0.32$ |
| w/o NU & $W_2$ | $17.68 \pm 0.25$ | $19.67 \pm 0.34$ | $23.09 \pm 1.97$ | $46.97 \pm 0.60$ | $34.16 \pm 0.23$ |

Table 8: Time taken (in seconds) training each model. The *ratio* is between the run-time of JSGNN and that of HGCN plus GCN.

| Method | Cora | Citeseer | Pubmed |
|---|---|---|---|
| GCN | 1.08 | 1.07 | 1.10 |
| HGCN | 3.87 | 3.96 | 4.44 |
| JSGNN | 18.53 | 21.60 | 24.73 |
| *Ratio* | 3.74 | 4.29 | 4.45 |

additional loss terms, JSGNN incurs only a small increase in time compared to its simpler variants across the evaluated datasets.

Additionally, in Table 8, we show the empirical run-time comparison. We see that the run-time ratio between JSGNN and HGCN plus GCN remains stable for graphs of different sizes. Therefore, the scalability of the well-studied models GCN, and HGCN implies that of JSGNN.

We show in Table 9 the model size comparison. We see that our model size is comparable to all the other hybrid models and approximately double those of the single-type models.

## C  Distributions of hyperbolicity

We include Table 10 that tabulates the percentage of nodes whose geometric hyperbolicity $\delta_v$ is (approximately) the prescribed value. Here, the hyperbolicity is taken in a neighborhood of each node to account for the local nature of message passing. We see, for example, for the Cora dataset, there are more than $52.4\%$ of nodes whose $\delta_v$ is at least 1. Hence, an Euclidean model is preferred. On the other hand, there are $\approx 23.0\%$ of nodes whose neighborhood is tree-like, and this is why a hybrid model can further improve the performance.

## D  Proof of Proposition 1

*Proof.* We first consider $\delta_{G,\infty}$. For two graphs $G_1 = (V_1, E_1)$ and $G_2 = (V_2, E_2)$, let $f_1 : G_1 \to M$, $f_2 : G_2 \to M$ be isometeric embeddings into a metric space $(M, d)$ such that $d_{GH}(G_1, G_2) = d_H(f_1(G_1), f_2(G_2))$. Denote $d_{GH}(G_1, G_2)$ by $\eta$. For $x, y, z, t$ in $G_1$, there are $x', y', z', t' \in G_2$ such that $d(f_1(x), f_2(x'))$, $d(f_1(y), f_2(y'))$,

Table 9: Number of trainable parameters.

| Method | Cora | Citeseer | Pubmed | CS | Photo |
|---|---|---|---|---|---|
| JSGNN (GAT+HGAT) | 47128 | 119696 | 17720 | 219967 | 25928 |
| $\kappa$-GCN | 46112 | 118720 | 16128 | 218272 | 24128 |
| GIL | 46700 | 119286 | 19155 | 219036 | 25380 |
| GAT | 23452 | 59752 | 8843 | 109244 | 12144 |
| GCN | 23335 | 59638 | 8339 | 109151 | 12072 |
| HGNN | 23335 | 59638 | 8339 | 109151 | 12072 |
| HGCN | 23336 | 59638 | 8339 | 109151 | 12072 |
| HGAT | 23431 | 59734 | 8435 | 109199 | 12120 |

Table 10: Hyperbolicity distribution in each dataset computed based on each node's two-hop subgraph. The lower the hyperbolicity, the more hyperbolic the node's neighborhood is.

| Dataset | 0 | 0.5 | 1 | 1.5 | 2 | 2.5 |
|---------|--------|--------|--------|--------|--------|--------|
| Disease | 1.0000 | 0.0000 | 0.0000 | 0.0000 | 0.0000 | 0.0000 |
| Cora | 0.2296 | 0.2230 | 0.5240 | 0.0225 | 0.0007 | 0.0000 |
| Citeseer | 0.5648 | 0.1350 | 0.2958 | 0.0045 | 0.0000 | 0.0000 |
| Pubmed | 0.3064 | 0.2225 | 0.3880 | 0.0814 | 0.0017 | 0.0000 |
| Airport | 0.0471 | 0.2566 | 0.6963 | 0.0000 | 0.0000 | 0.0000 |
| Photo | 0.0251 | 0.0482 | 0.8825 | 0.0442 | 0.0000 | 0.0000 |
| CS | 0.0657 | 0.1284 | 0.5718 | 0.2332 | 0.0009 | 0.0000 |

$d(f_1(z), f_2(z'))$, $d(f_1(t), f_2(t'))$ are all bounded by $\eta$. We now estimate:

$$
\begin{aligned}
d_{G_1}(x, y) + d_{G_1}(z, t) &= d(f_1(x), f_1(y)) + d(f_1(z), f_1(t)) \\
&\leq d(f_2(x'), f_2(y')) + d(f_2(z'), f_2(t')) + 4\eta \\
&= d_{G_2}(x', y') + d_{G_1}(z', t') + 4\eta \\
&\leq \max\{d_{G_2}(x', z') + d_{G_2}(y', t'), d_{G_2}(z', y') + d_{G_2}(x', t')\} \\
&\quad + 2\delta_{G_2, \infty} + 4\eta \\
&\leq \max\{d(f_1(x), f_1(z)) + d(f_1(y), f_1(t)), \\
&\quad d(f_1(z), f_1(y)) + d(f_1(x), f_1(t))\} \\
&\quad + 2\delta_{G_2, \infty} + 8\eta \\
&= \max\{d_{G_1}(x, z) + d_{G_1}(y, t), d_{G_1}(z, y) + d_{G_1}(x, t)\} \\
&\quad + 2\delta_{G_2, \infty} + 8\eta.
\end{aligned}
\tag{19}
$$

Therefore, $\delta_{G_1, \infty} \leq \delta_{G_2, \infty} + 4\eta$. By the same argument swapping the role of $G_1$ and $G_2$, we have $\delta_{G_2, \infty} \leq \delta_{G_1, \infty} + 4\eta$. Therefore $|\delta_{G_1, \infty} - \delta_{G_2, \infty}| \leq 4\eta$ and $\delta_{G, \infty}$ is Lipschitz continuous w.r.t. $G$.

The proof of the continuity of $\delta_{G, 1}$ is more involved. Consider $G_1$ and $G_2$ in $\mathcal{G}_\epsilon$. Let $f_1, f_2, (M, d), \eta$ be as earlier and assume $\eta \ll \epsilon$, for example, $\eta = \alpha \epsilon$ for $\alpha$ is smaller than all the numerical constants in the rest of the proof.

We adopt the following convention: for any non-vertex point of a graph, its degree is 2. By subdividing the edges of $G_1$ and $G_2$ if necessary, we may assume that the length of each edge $e$ in $E_1$ or $E_2$ satisfies $\epsilon/2 \leq e < \epsilon$. As a consequence, for $(u, v)$ in $E_1$ (resp. $E_2$), $d_{G_1}(u, v)$ (resp. $d_{G_2}(u, v)$) is the same as the length of $(u, v)$. We define a map $\phi : G_1 \to G_2$ as follows. For $v \in G_1$, there is a $v'$ in $G_2$ such that $d_{GH}(f_1(v), f_2(v')) \leq \eta$. Then we set $\phi(v) = v'$. The map $\phi$ is injective on the vertex set $V_1$. Indeed, for $u \neq v \in V_1$, $d_{G_1}(u, v) \geq \epsilon/2$ and hence $d_{G_2}(\phi(u), \phi(v)) \geq \epsilon/2 - 2\eta > 0$. The strategy is to modify $\phi$ by a small perturbation such that the resulting function $\psi : G_1 \to G_2$ is a homeomorphism that is almost an isometry.

For $v \in V_1$, let $N_v$ be the $5\eta$ neighborhood of $v$. It is a star graph and its number of branches is the same as the degree of $v$, say $k$. Let $v_1, \ldots, v_k$ be the endpoints of $N_v$. The convex hull (of shortest paths) $C_v$ of $\{\phi(v_1), \ldots, \phi(v_k)\}$ in $G_2$ is also a star graph. This is because $C_v$ is contained in the $7\eta$ neighborhood of $\phi(v)$ and it contains at most 1 vertex in $V_2$.

We claim that $C_v$ has the same number of branches as $N_v$. First of all, $C_v$ cannot have fewer branches. For otherwise, there is a $\phi(v_i)$ in the path connecting $\phi(v)$ and $\phi(v_j)$ for some $j \neq i$. Hence,

$$
\begin{aligned}
d_{G_2}(\phi(v_i), \phi(v_j)) &\leq d_{G_2}(\phi(v_i), \phi(v)) \leq 7\eta \\
&< 10\eta - 2\eta = d_{G_1}(v_i, v_j) - 2\eta.
\end{aligned}
$$

This is a contradiction with the property of $\phi$. It cannot have more branches than $k$ as it is the convex hull of at most $k$ points.

We next consider different cases for $k$. For $k \neq 2$, as $C_v$ is a star graph, it has a unique node $v'$ with degree $k$ (in $C_v$), and $d_{G_1}(v', \phi(v_j)) > 0, 1 \leq j \leq j$. We claim that $v'$ has degree exact $k$ in $G_2$. Suppose on the contrary, its degree in $G_2$ is larger than $k$. Then there is a branch not contained in $C_v$. Let $w'$ be a node on the new branch such that $6\eta \leq d_{G_2}(w', \phi(v)) \leq 7\eta$. Moreover, there is a node $w$ in $N_v$ such that $4\eta \leq d_{G_1}(w, v) \leq 9\eta$ and $w' = \phi(w)$. Moreover, $w$ is on the branch containing $v_j$ for some $j$, and hence $d_{G_1}(w, v_j) \leq 4\eta$. Therefore,

$$\begin{aligned}
d_{G_1}(w, v_j) &\leq 6\eta - 2\eta \\
&< d_{G_2}(v', \phi(v_j)) + d_{G_2}(w', v') - 2\eta \\
&= d_{G_2}(\phi(w), \phi(v_j)) - 2\eta,
\end{aligned}$$

which is a contradiction. In this case, we define $\psi(v) = v' \in G_2$. If $k = 2$ when $N_v$ is a path, by a similar argument, we have that $C_v$ is a path. We set $\psi(v) = \phi(v)$. An illustration is given in Fig. 7.

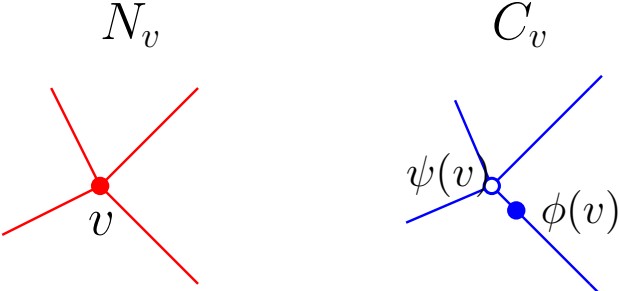

Figure 7: Illustration of $\psi$.

For each $v \in V_1$, we now enlarge the neighborhood and consider its $\epsilon/6$-neighborhood $N_v'$. It does not contain another vertex and hence is also a star graph. Moreover, if $v \neq u \in V_1$, then $N_v' \cap N_u' = \emptyset$ for otherwise $d_{G_1}(u, v) \leq \epsilon/3$, which is impossible. We may similarly consider the $\epsilon/6$-neighborhoods $C_u', C_v'$ of $\psi(u)$ and $\psi(v)$. Both $C_u'$ and $C_v'$ do not contain any vertex in $V_2$ with degree $\neq 2$.

As $N_v'$ and $C_v'$ are star graphs with the same number of branches, there is an isometry (also denoted by) $\psi : N_v' \to C_v'$ such that $d_{G_2}(\psi(w), \phi(w)) \leq 2\eta$. By disjointedness of $\epsilon/6$ neighborhoods, we may combine all the maps above together to obtain $\psi : \cup_{v \in V_1} N_v' \to \cup_{v \in V_1} C_v'$.

For the rest of $G_1$, consider any edge $(u, v) \in E_1$. Without loss of generality, let $u_1$ and $v_1$ be the leaves of $N_u'$ and $N_v'$ contained in $(u, v)$. We claim that the shortest open path connecting $\psi(u_1)$ and $\psi(v_1)$ is disjoint from $\cup_{v \in V_1} C_v'$. For otherwise, $d_{G_1}(u_1, v_1) \geq 2\epsilon/3$, while $d_{G_2}(\phi(u_1), \phi(u_2)) \leq d_{G_2}(\psi(u_1), \psi(u_2)) - 4\eta \geq \epsilon/2 + 2\epsilon/6 - 4\eta$. Therefore, $2\epsilon/3 - 2\eta \geq 5\epsilon/6 - 4\eta$, which is impossible as $\eta \ll \epsilon$.

Let $P_{u,v}$ and $Q_{u,v}$ be the shortest paths connecting $u_1, v_1$ and $\psi(u_1), \psi(v_1)$ respectively (illustrated in Fig. 8). Then the length of $P_{u,v}$ and $Q_{u,v}$ differ at most by $4\eta$. We may further extend $\psi : P_{u,v} \to Q_{u,v}$ by a linear scaling such that $d_{G_2}(\psi(w), \phi(w)) \leq 3\eta$ for $w \in P_{u,v}$. For different edges $(u, v), (u', v')$, it is apparent $Q_{u,v} \cap Q_{u',v'}$ are disjoint, as the minimal distance between points on $P_{u,v}$ and $P_{u',v'}$ is at least $\epsilon/3$. Therefore, we obtain a continuous injection $\psi : G_1 \to G_2$, which maps homeomorphically onto its image.

We claim that $\psi$ is onto. If not, there is a vertex $v' \in V_2$ that is not in $\psi(V_1)$ but it has a neighboring vertex $u' = \psi(u)$. However, this implies that the degree of $u'$ is strictly larger than that of $u$, which is impossible as we have shown.

In summary, $\psi : G_1 \to G_2$ is a homeomorphism such that $|d_{G_1}(u, v) - d_{G_2}(u, v)| \leq 6\eta$ for any $u, v \in G_1$. Moreover, $\psi$ is piecewise linear whose gradient $\psi'$ is 1 in the interior of $N_v', v \in V_1$ and satisfies

$$\frac{\frac{\epsilon}{6} - 6\eta}{\frac{\epsilon}{6}} \leq \psi'(w) \leq \frac{\frac{\epsilon}{6} + 6\eta}{\frac{\epsilon}{6}}, \tag{20}$$

for $w$ contained in the interior of some $P_{u,v}, (u, v) \in E_1$.

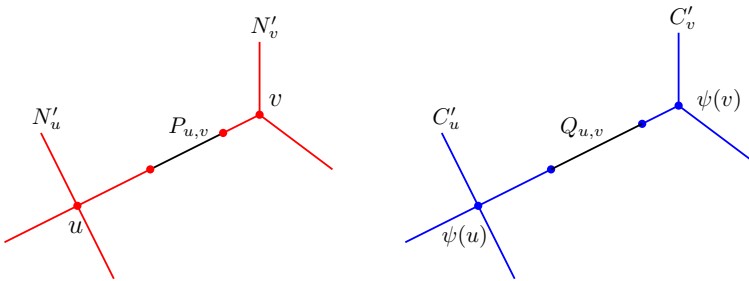

Figure 8: Illustration of $P_{u,v}$ and $Q_{u,v}$.

We are ready to estimate $|\delta_{G_1,1} - \delta_{G_2,1}|$. Let $|G_i|$ be the total edge weights of $G_i, i = 1, 2$. For convenience, we denote a typical tuple $(u, v, w, t) \in G_1^4$ as a vector $\mathbf{v}$, and $(\psi(u), \psi(v), \psi(w), \psi(t))$ by $\boldsymbol{\psi}(\mathbf{v})$. The map $\boldsymbol{\psi} : G_1^4 \to G_2^4, \mathbf{v} \mapsto \boldsymbol{\psi}(\mathbf{v})$ inherits the properties of its counterpart $\psi$, which is a piecewise linear homeomorphism. In particular, its Jacobian $J(\mathbf{v})$ is defined almost everywhere. Using Definition 2, we have:

$$
\begin{aligned}
&|\delta_{G_1,1} - \delta_{G_2,1}| \\
&= \left| \int_{\mathbf{v} \in G_1^4} |G_1|^{-4} \tau_{G_1}(\mathbf{v}) \, d\mathbf{v} \right. \\
&\quad \left. - \int_{\mathbf{v} \in G_1^4} |G_2|^{-4} J(\mathbf{v}) \tau_{G_2}(\boldsymbol{\psi}(\mathbf{v})) \, d\mathbf{v} \right| \\
&\leq \sup_{\mathbf{v} \in G_1^4} \left| \tau_{G_1}(\mathbf{v}) - \frac{|G_1|^4}{|G_2|^4} J(\mathbf{v}) \tau_{G_2}(\boldsymbol{\psi}(\mathbf{v})) \right|.
\end{aligned}
\tag{21}
$$

Similar to (19), we estimate

$$
\sup_{\mathbf{v} \in G_1^4} |\tau_{G_1}(\mathbf{v}) - \tau_{G_2}(\boldsymbol{\psi}(\mathbf{v}))| \leq 24\eta.
\tag{22}
$$

Moreover, we have seen in the proof that $\psi$ can only have distortion when restricted to $P_{u,v}$ for $(u, v) \in E_1$. As

$$
\frac{\frac{2\epsilon}{3} - 6\eta}{\frac{2\epsilon}{3}} \leq |P_{u,v}|/|Q_{u,v}| \leq \frac{\frac{2\epsilon}{3} + 6\eta}{\frac{2\epsilon}{3}},
$$

the same bounds holds for $|G_1|/|G_2|$. Both upper and lower bounds can be arbitrarily close to 1 if $\eta$ is small enough. Similarly, by (20), $J(\mathbf{v})$ as a fourth power of $\psi'$ can also be made arbitrarily close to 1. In conjunction with (21) and (22), $|\delta_{G_1,1} - \delta_{G_2,1}|$ can be arbitrarily small if $\eta$ is chosen to be small enough. This proves that $\delta_{G,1}$ is continuous in $G$. $\qquad \square$

## E   Related works

In this appendix, models that utilize multiple spaces and advanced topological information such as curvature are reviewed.

CurvGN (Ye et al., 2020) and Curvature Graph Neural Network (CGNN) (Li et al., 2021) learn to reweigh messages propagated between nodes, in Euclidean space, using curvature information. Curvature measures how easily information flows between two nodes. These works assume that the edges with low curvatures indicate the class boundaries, thus low weights are assigned when the edges are of low curvature. In our work, we use Gromov hyperbolicity instead of Ollivier's Ricci curvature to choose the appropriate space and we do not reweigh the edges.

To the best of our knowledge, the closest works to ours are Geometry Interaction Learning (GIL) (Zhu et al., 2020) and $\kappa$-GCN (Bachmann et al., 2019). $\kappa$-GCN utilizes the (Cartesian) product space to model data

in different spaces and employs the $\kappa$-stereographic model in each of the spaces. The Cartesian product enables a combinatorial construction of the mixed curvature space, thus the representations are first learned independently in the respective spaces and then concatenated (Sun et al., 2022). In terms of implementation, this is similar to our framework but instead of concatenation, we introduce a space selection mechanism guided by hyperbolicity to fuse the representations.

On the other hand, GIL proposes a feature interaction scheme to leverage different spaces and a probability assembling module to combine the classification probabilities for obtaining the final prediction. The feature interaction scheme is where the node features in the respective two spaces are enhanced based on the distance similarity of the two sets of spatial embeddings. The larger the distance between the different spatial embeddings, the more significant the portion of features from the other space is summed to itself as seen in Fig. 3. However, this may introduce noise to the respective spaces, which we explain further below.

Our approach differs from GIL (Zhu et al., 2020) in some key aspects. Firstly, we leverage the distribution of geometric hyperbolicity to guide our model to learn to decide if each node better embedded in a Euclidean or hyperbolic space instead of performing feature interaction learning. This is done by aligning the distribution of the learned model hyperbolicity and geometric hyperbolicity using the Wasserstein distance. Our motivation is that if a node can be best embedded in one of the two spaces, encoding it in another space other than the optimal one would result in comparably larger distortions. Minimal information would be present in the sub-optimal space to help "enhance" the representation in the better space. Hence, promoting feature interaction could possibly introduce more noise to the corresponding spaces. The ideal situation is then to learn normalized selection weights that are non-uniform for each node so that we select for each node a single, comparably better space's output embedding. To achieve this, we introduce an additional loss term that promotes non-uniformity. Lastly, we do not require probability assembling since we only have one set of output features at the end of the selection process.

## F   Limitations

The paper does not have a comprehensive numerical study on extremely large datasets. Therefore, we do not have a definite answer to how JSGNN will perform on these graphs. However, the datasets being studied in our paper are diverse enough in terms of geometric properties. We have theoretically analyzed the complexity, which depends on the complexities of the hyperbolic and Euclidean base models. Therefore, its scalability also depends on the based models. For example, GCN, GAT, HGCN, and HGAT all scale well with graph size. Notice that message passing is a local operation, and hence model performance on graphs of medium sizes should be illustrative enough.

In future works, it is worthwhile to test our model on extremely large graphs to verify whether the above speculation holds.

## G   Broader impact

Since this research focuses on graph neural networks applied to publicly available datasets, there is no immediate broader impact concerns. The datasets used are standard benchmarks in the field, and there is no indication that the work introduces any significant ethical risks.

