# OpenReview forum: "Node-Specific Space Selection via Localized Geometric Hyperbolicity in Graph Neural Networks"
_TMLR — Accepted by TMLR_

### Review · Reviewer_mCbs · 2024-09-21

**Summary Of Contributions:**

This paper starts by calling out that real-world graphs are complex which could be made of both euclidean and hyperbolic structures, therefore explores the idea of integrating both Euclidean and hyperbolic space for node representation based on the hyperbolicity of local nodes in a graph neural network. The authors introduced JSGNN, which is a joint space graph neural network where dual-space embeddings are combined through attention mechanism. To optimize this model, distribution match is conducted between the geometric hyperbolicity of local graph structure and the model hyperbolicity of the GNN, ensuring that both Euclidean and hyperbolic embeddings are available where needed in the graph. The authors also empirically prove through different tasks on multiple benchmark datasets that JSGNN demonstrated competitive performance.

**Audience:**

Yes

**Broader Impact Concerns:**

The broader impact of this paper has not been clearly justified yet.

**Claims And Evidence:**

Yes

**Requested Changes:**

• Discuss the computation complexities of nodes representation in JSGNN and how it grows with larger scaled dataset

• A description of the benchmark datasets on hyperbolic and euclidean features

**Strengths And Weaknesses:**

Strengths:

• The necessary explanations and background provided as to why solely GAT or hyperbolic GAT are not ideal for complex graph, justifying the opportunity and motivation of this paper. It’s a novel idea to combine Euclidean and hyperbolic geometries for each node representation

• The experiments on multiple datasets are informative and convincing. The results show JSGNN exceeds the performance of single-space baselines on most datasets and demonstrate the benefit of including both euclidean spaces and hyperbolic spaces

• The ablation study isolates and studies the impact of individual components (non-uniformity constraint, Wasserstein metric) and provides insightful analysis into their contributions.

• The thoroughness of including Non-uniformity of selection weights in the loss function to deviate attention weights from 0.5 and maximize the use of JSGNN


Weaknesses:

Overall the paper is well-written and convincing, with enough theories and experiments backing up. Below are a few points where the paper can be improved

• In node classification part, I didn’t see the dataset characteristics description provided to explain why hyperbolic models work well for certain datasets, Euclidean models work well for the others, and why JSGNN can exceed the performance on most datasets.

• It would be beneficial to discuss the tradeoff between computational costs of JSGNN and performance as model scales with larger datasets

---

> ### Author Response · Authors · 2024-09-30
> **Response to the reviewer's comments (part 1)**
>
> We would like to thank the reviewer for taking the time to review our paper and his/her helpful comments. Below, we provide our point-to-point responses.
>
> Comment 1: (From weaknesses) In node classification part, I didn’t see the dataset characteristics description provided to explain why hyperbolic models work well for certain datasets, Euclidean models work well for the others, and why JSGNN can exceed the performance on most datasets. (From Request Changes) A description of the benchmark datasets on hyperbolic and euclidean features.
>
> Response 1:     When people refer to a dataset as being hyperbolic or Euclidean, usually the hyperbolicity of the entire graph is estimated. However, a graph neural network based message passing usually aggregates features from a small neighborhood $N_v$ of each node $v$. Therefore, the hyperbolicity $\delta_v$ of every $N_v$ (called the geometric hyperbolicity in the paper), instead of the hyperbolicity of the entire graph, is a more important indicator regarding whether we should use an Euclidean or a hyperbolic model. For example, though Cora is usually regarded as an Euclidean dataset, there is a large percentage of nodes having a neighborhood with small geometric hyperbolicity $\delta_v$. As $\delta_v$ varies from node to node, JSGNN, being a hybrid model that takes this into account, can exceed the performance of models belonging to a single type on most datasets.
>
> As suggested by the reviewer, in the revision, we shall include the following table that tabulates the percentage of nodes whose geometric hyperbolicity $\delta_v$ is (approximately) the prescribed value. We see, for example, for the Cora dataset, there are approximately 50% of nodes whose $\delta_v$ is at least $1$. Hence, an Euclidean model is preferred. On the other hand, there are $\approx$23.7% of nodes whose neighborhood is tree-like, and this is why a hybrid model can further improve the performance.
> | Dataset   | 0      | 0.5    | 1      | 1.5    | 2      | 2.5    |
> |-----------|--------|--------|--------|--------|--------|--------|
> | Disease   | 1.0000 | 0.0000 | 0.0000 | 0.0000 | 0.0000 | 0.0000 |
> | Cora      | 0.2371 | 0.2622 | 0.4934 | 0.0074 | 0.0000 | 0.0000 |
> | Citeseer  | 0.5648 | 0.1350 | 0.2958 | 0.0045 | 0.0000 | 0.0000 |
> | Pubmed    | 0.3222 | 0.2398 | 0.4189 | 0.0189 | 0.0003 | 0.0000 |
> | Airport   | 0.0471 | 0.2566 | 0.6963 | 0.0000 | 0.0000 | 0.0000 |
> | Photo     | 0.0251 | 0.0482 | 0.8825 | 0.0442 | 0.0000 | 0.0000 |
> | CS        | 0.0662 | 0.1709 | 0.6993 | 0.0635 | 0.0001 | 0.0000 |

---

> ### Author Response · Authors · 2024-09-30
> **Response to the reviewer's comments (part 2)**
>
> Comment 2: (From weaknesses) It would be beneficial to discuss the tradeoff between computational costs of JSGNN and performance as model scales with larger datasets. (From Requested Changes) Discuss the computation complexities of nodes representation in JSGNN and how it grows with larger scaled dataset.
>
> Response 2:  As we have explained in Appendix B: Complexity, our model has two essential parts.
>
> We first incur an overhead to compute the geometric hyperbolicity $\delta_v$ for each node. We follow and modify the implementation of [1] rather than using the naive implementation with a time complexity of $O(N^4)$, $N=|V|$. Instead of considering all possible 4-tuples in the graph, we heuristically sample $K$ 4-tuples from each nodes' two-hop subgraph. Consequently, the complexity is $O(N) \times [O(K \times E_{subgraph})+ O(E_{subgraph})] \approx O(N(K+1)d^2)$, where $d$ denotes the average degree in the graph, $K$ denotes the number of samples. $E_{subgraph}$ represents the number of edges in the two-hop subgraph of each node, which is approximately $O(d^2)$. The term $O(K \times E_{subgraph})$ is the complexity to compute $\delta_v$ and the extra $O(E_{subgraph})$ is the complexity for finding the local neighborhood.
>
> Suppose the graph is sparse, which is true for most datasets being studied in the paper. Then $d$ is small. This runtime can be further optimized by parallelization, as each node's calculation is independent. Also, notice that this step is performed only once during pre-processing, and the step is model-agnostic. For a dataset, the computed $\{\delta_v\}_{v\in V}$ can be stored and re-used for different trainings and also for different base models. Moreover, if the dataset is updated, we only need to adjust $\delta_v$ for nodes $v$ whose neighborhood structures are changed during the update, which can be done efficiently.
>
> For training of the model, we usually use two base models $M_{e}$ (e.g., GCN) and $M_{h}$ (e.g., HGCN) for handling Euclidean and hyperbolic structures respectively. The complexity of the message passing in our model is $O(C_e+C_h)$, where $C_e$ and $C_h$ are the message passing complexities of $M_e$ and $M_h$ respectively.
>
> Regarding the implementation of the extra loss terms, calculating the Wasserstein distance for one-dimensional distributions requires a time complexity of $O(N\log N)$ where $N$ is the number of nodes in the graph, primarily dominated by the time needed to sort the distributions. Meanwhile, obtaining the non-uniformity loss has a time complexity of $O(N)$.
>
> In the revision, we shall update Appendix B: Complexity with more details as described above.
>
> [1] Ines Chami, Zhitao Ying, Christopher Ré, and Jure Leskovec. Hyperbolic graph convolutional neural networks. In Advances in Neural Information Processing Systems, pp. 4869–4880, 2019. (https://github.com/HazyResearch/hgcn/blob/master/utils/hyperbolicity.py)

---

### Review · Reviewer_RicP · 2024-09-23

**Summary Of Contributions:**

The paper introduces a novel approach to embedding nodes in Graph Neural Networks (GNNs) by leveraging both Euclidean and hyperbolic spaces. The authors propose that real-world graphs often contain regions with mixed geometrical properties, making it inefficient to embed all nodes into a single geometric space. To address this, they introduce two forms of localized hyperbolicity—geometric (Gromov) and model-based—that help determine the most suitable space for each node. Their proposed Joint Space Graph Neural Network (JSGNN) architecture learns node embeddings in both Euclidean and hyperbolic spaces simultaneously, using an attention mechanism guided by the Wasserstein metric to select the optimal embedding space for each node. Additionally, the authors introduce a non-uniformity loss to encourage the model to select either Euclidean or hyperbolic space for each node, avoiding distortions caused by partial embeddings in both spaces. Through evaluations on node classification and link prediction tasks across multiple benchmark datasets, JSGNN demonstrates superior performance compared to baseline models by effectively leveraging both Euclidean and hyperbolic geometries in graph learning.

**Audience:**

Yes

**Broader Impact Concerns:**

Since this research focuses on graph neural networks applied to publicly available datasets, I do not foresee any immediate broader impact concerns. The datasets used are standard benchmarks in the field, and there is no indication that the work introduces any significant ethical risks.

**Claims And Evidence:**

Yes

**Requested Changes:**

1. Some citations are not properly formatted. The authors are advised to carefully check and correct all instances of \cite and \citep to ensure consistent and accurate referencing.

2. In the definition of $ \mathbb{D}^n_c$, please explicitly specify the type of norm used for clarity and to avoid ambiguity in the interpretation of the mathematical space.

3. In Definition 2, equation (10) presents an inequality. It is unclear how the authors achieve $\inf_{\delta>0} \{(10)\}$. Please provide additional explanation or steps to clarify this derivation.

4. Figure 3 could be made clearer. It is recommended that the authors improve the clarity of the figure by using higher resolution graphics or adjusting the visual elements to make the relationships in the diagram easier to understand.

5. Provide a more detailed analysis of the computational complexity of the proposed model, particularly the pre-processing step of calculating geometric hyperbolicity. This analysis should include time and memory complexity, as well as scalability considerations for large graphs. This is crucial for assessing the model’s practicality in real-world applications.

**Strengths And Weaknesses:**

Strengths:

- The introduction of node-specific space selection in GNNs through localized geometric hyperbolicity is an innovative approach that addresses the limitations of using a single geometric space for the entire graph. This allows for better handling of complex graphs with mixed geometrical properties.

- The authors provide a thorough evaluation of their model, including ablation studies that highlight the importance of key components like non-uniformity loss and Wasserstein alignment. This provides useful insights into the functioning and benefits of the proposed architecture.

- The paper provides strong theoretical backing for the proposed model, including detailed explanations of geometric and model-based hyperbolicity. The use of the Wasserstein metric to align these hyperbolicities adds mathematical rigor to the space selection process.


Weaknesses:

- The paper mentions the computational cost of calculating geometric hyperbolicity but lacks a detailed analysis of how this affects scalability, particularly for large graphs. A more thorough discussion of the model's computational overhead would be useful for practical implementations.

- There is limited discussion regarding potential limitations of the model. For instance, the model may not perform as well on graphs that are either entirely Euclidean or entirely hyperbolic, and this issue is not fully explored. Additionally, the trade-offs between performance and computational complexity are not sufficiently discussed.

- The dual-space architecture and the additional loss terms (non-uniformity loss and Wasserstein alignment) add complexity to the model. Although the empirical results justify these components, the authors could discuss whether simpler variants of the model might still achieve competitive results with reduced computational burden.

---

> ### Author Response · Authors · 2024-10-09
> **Response to the reviewer's comments (part 1)**
>
> We would like to thank the reviewer for taking the time to review our paper and his/her helpful comments. Below, we provide our point-to-point responses.
>
> Weaknesses 1: The paper mentions the computational cost of calculating geometric hyperbolicity but lacks a detailed analysis of how this affects scalability, particularly for large graphs. A more thorough discussion of the model's computational overhead would be useful for practical implementations.
>
> Requested Change 5: Provide a more detailed analysis of the computational complexity of the proposed model, particularly the pre-processing step of calculating geometric hyperbolicity. This analysis should include time and memory complexity, as well as scalability considerations for large graphs. This is crucial for assessing the model’s practicality in real-world applications
>
> Response: For the complexity of the overhead that computes the geometric hyperbolicity, we need to compute $\delta_v$ for each $v\in V$. We follow and modify the implementation of [1] rather than using the naive implementation with a time complexity of $O(N^4)$, $N=|V|$. Instead of considering all possible 4-tuples in the graph, we heuristically sample $K$ 4-tuples from each nodes' two-hop subgraph. Consequently, the complexity is $O(N) \times [O(K \times E_{subgraph}) + O(E_{subgraph}))] \approx O(N(K+1)d^2)$, where $d$ denotes the average degree in the graph, $K$ denotes the number of samples. $E_{subgraph}$ represents the average number of edges in the two-hop subgraph of each node, which is approximately $d^2$. The term $O(E_{subgraph})$ is the complexity of identifying a $2$-hop neighborhood, and $O(K \times E_{subgraph})$ is the complexity of computing $\delta_v$.
>
> Assuming the graph is sparse, $d$ is small. This runtime can be further optimized by parallelization, as each node's calculation is independent. Also, notice that this step is performed only once during pre-processing, and the step is model-agnostic. For a dataset, the computed {$\delta_v\, v\in V$} can be stored and re-used for different trainings and also for different base models. Moreover, if the dataset is updated, we only need to adjust $\delta_v$ for nodes $v$ whose neighborhood structures are changed during the update, which can be done efficiently. However, the complexity may grow quickly for dense graphs.
>
> For the memory complexity, we store the value $\delta_v$ for each node. Therefore, the memory complexity is $O(|V|)$. This is minimal compared to the memory complexity of storing the node features $O(V\times F)$ where $F$ denotes feature size, the graph $O(V^2)$ or $O(V+E)$ depending on the storage format of adjacency matrix or adjacency list. Additionally, we have the weight matrices to transform the input features, which require $O(2 \times L \times F \times F')$ where $F'$ is the hidden dimension and $L$ is the number of layers. The constant $2$ accounts for the two GNN components: one in Euclidean space and the other in hyperbolic space. Furthermore, there are learnable weights in the form of attention weights.
>
> For training of the model, we usually use two base models $M_{e}$ (e.g., GCN) and $M_{h}$ (e.g., HGCN) for handling Euclidean and hyperbolic structures respectively. The complexity of the message passing in our model is $O(C_e+C_h)$, where $C_e$ and $C_h$ are the message passing complexities of $M_e$ and $M_h$ respectively.
>
> We shall include the above discussion in the revision to give readers a clearer idea.
>
> [1] Ines Chami, Zhitao Ying, Christopher Ré, and Jure Leskovec. Hyperbolic graph convolutional neural networks. In Advances in Neural Information Processing Systems, pp. 4869–4880, 2019. https://github.com/HazyResearch/hgcn/blob/master/utils/hyperbolicity.py

---

> ### Author Response · Authors · 2024-10-09
> **Response to the reviewer's comments (part 2)**
>
> Weaknesses 2: There is limited discussion regarding potential limitations of the model. For instance, the model may not perform as well on graphs that are either entirely Euclidean or entirely hyperbolic, and this issue is not fully explored. Additionally, the trade-offs between performance and computational complexity are not sufficiently discussed.
>
> Response: We agree with the reviewer that our model may underperform in extreme cases. For example, the graph of the Disease dataset is a tree and thus entirely hyperbolic. Two models dedicated to hyperbolic datasets slightly outperform our hybrid model for the node classification task (see Table 2). This is the compensation for considering a hybrid model as it is unlikely that the soft matching of geometric and model hyperbolicity drives the model entirely hyperbolic. However, our model remains competitive for most other datasets and the link prediction task.
>
> As suggested, we shall include the above discussion when we revise the paper.
>
> Regarding the computation cost, we refer the reviewer to our response to Weaknesses 1 and Weaknesses 3 for the details.
>
> Weaknesses 3: The dual-space architecture and the additional loss terms (non-uniformity loss and Wasserstein alignment) add complexity to the model. Although the empirical results justify these components, the authors could discuss whether simpler variants of the model might still achieve competitive results with reduced computational burden.
>
> Response: The performance of the simpler models is analyzed in Table 4 (ablation study). In the revision, we provide the following table regarding the time taken for simple models in the format analogous to Table 4. Time taken (in seconds) for 300 epochs for node classification task is shown. We see that the "full model" does not incur much additional computation cost as compared with its simpler versions.
> | Method                | Cora              | Citeseer          | Pubmed            | CS                | Photo            |
> |-----------------------|-------------------|-------------------|-------------------|-------------------|------------------|
> | JSGNN (GAT+HGAT)       | 18.53 ± 1.05       | 21.60 ± 0.62       | 24.73 ± 3.49       | 49.35 ± 0.62       | 37.03 ± 0.51      |
> | w/o W_2               | 17.29 ± 0.66       | 20.06 ± 0.27       | 23.56 ± 1.85       | 47.31 ± 0.34       | 34.45 ± 0.41      |
> | w/o NU                | 18.06 ± 0.13       | 19.98 ± 0.32       | 23.42 ± 3.49       | 48.36 ± 0.41       | 36.71 ± 0.32      |
> | w/o NU & W_2          | 17.68 ± 0.25       | 19.67 ± 0.34       | 23.09 ± 1.97       | 46.97 ± 0.60       | 34.16 ± 0.23      |
>
> Requested Changes 1: Some citations are not properly formatted. The authors are advised to carefully check and correct all instances of \cite and \citep to ensure consistent and accurate referencing.
>
> Response: In the revision, we shall carefully check all the citations as suggested to ensure they are properly formatted.
>
> Requested Changes 2: In the definition of $\mathbb{D}_c^n$, please explicitly specify the type of norm used for clarity and to avoid ambiguity in the interpretation of the mathematical space.
>
> Response: In the Definition $\mathbb{D}_c^n = \\{ {\bf x} \in \mathbb{R}^n : c\||{\bf x}\|| < 1 \\}$, the norm $||\cdot||$ here is the usually Euclidean norm. This is the reason why the model is called the Poincare ball model. However, the metric on $\mathbb{D}_c^n$ is not the Euclidean metric. In the revision, we shall explain the terms clearly as above.
>
> Requested Changes 3: In Definition 2, equation (10) presents an inequality. It is unclear how the authors achieve $\inf_{\delta\geq 0}$. Please provide additional explanation or steps to clarify this derivation.
>
> Response: First of all, it is required that $\delta\geq 0$, and notice that $\delta=0$ is allowed, e.g., $X$ is a tree. For a general compact metric space $X$, we consider the set $S_{x,y,z,t} = \\{\delta \geq 0 \mid (10) \text{ is satisfied for } x,y,z,t\\}$. Any $\delta\in S_{x,y,z,t}$ satisfies
>     \begin{align*}
>        \delta \geq \frac{1}{2}\big(d(x,y) + d(z,t) - \max\{d(x,z)+d(y,t),d(z,y)+d(x,t)\}\big).
>     \end{align*}
>     The set $S_{x,y,z,t}$ is clearly non-empty (as we can always choose $\delta$ to be very large). Therefore, in Definition 2, $\inf_{\delta\geq 0}$ is referring to the infimum of the set $S_{x,y,z,t}$.
>
> We shall include the above explanation when revising the paper.
>
> Requested Changes 4: Figure 3 could be made clearer. It is recommended that the authors improve the clarity of the figure by using higher resolution graphics or adjusting the visual elements to make the relationships in the diagram easier to understand.
>
> Response: In the revision, we shall follow the reviewer's suggestion to redraw Fig. 3 for better resolution and presentation.
>
> Requested Changes 5: See Weaknesses 1.

---

### Review · Reviewer_HXvf · 2024-10-08

**Summary Of Contributions:**

Considering that real-world graphs are a combination of hyperbolic and Euclidean geometries, this paper presents a new graph representation learning approach by integrating Euclidean and hyperbolic space representation learning, called the Joint Space Graph Neural Network (JSGNN). The model uses two notions of hyperbolicity—geometric and model-based—and aligns their distributions through the Wasserstein metric to determine optimal embeddings.


The claimed contributions of this paper are:
1. Exploring and analyzing two notions of local hyperbolicity—geometric and model-based—to describe the underlying local geometry and determine the preferred embedding space for each node.
2. Proposing the Joint Space Graph Neural Network (JSGNN), which leverages both Euclidean and hyperbolic spaces during learning by allowing node-specific geometry space selection.
3. Empirical evaluations show that JSGNN outperforms baseline models in node classification and link prediction tasks.

**Audience:**

Yes

**Broader Impact Concerns:**

I have no Broader Impact Concerns.

**Claims And Evidence:**

No

**Requested Changes:**

Please see **Weaknesses**

**Strengths And Weaknesses:**

**Strengths**
1. Combining Euclidean and hyperbolic spaces for node-specific embeddings would be a promising and reasonable way to handle heterogeneous graph geometries.
2. The motivation of the paper in Figure 1 is clear and makes sense to me.




**Weaknesses**
1. The experiments in Tables 1 and 2 are not fair to me. For example, JSGNN is a combination of GCN and HGCN, so the number of parameters will roughly double compared to GCN (correct me if I am wrong here). If this is the case, the results put the proposed method in an unfair position.
2. If the above is true, the marginal improvement in Table 1 can be ignored and is not promising at all.
3. While the motivating example in Figure 2 makes sense to me, this paper does not show that it is reflected in real graphs (despite the distributions of geometric hyperbolicity in Figure 2). This paper should also plot real graphs that include tree and lattice structures.
4. Figure 3 is of low quality and not professional to me.
5. As this paper claims that node representation is geometric (Gromov) and model-based, to determine the preferred embedding space for each node, the visualization of the node embeddings is a must-have experiment to show the learned node representations and compare them to hyperbolic and Euclidean representations.
6. Based on Weakness 1, this paper should include a discussion on the scalability of JSGNN and analyze the time complexity of JSGNN, particularly regarding computational costs compared to GNN and HGNN. Providing empirical results on how well the method scales with graph size would also be helpful.
7. I would also recommend expanding the experiments to include synthetic graph datasets like in Figure 1, especially where both Euclidean and hyperbolic structures are observed. This would help demonstrate the effectiveness of the proposed method.
8. The datasets used in the experiments are quite small at this time. The largest graph is Pubmed, which may limit the scope of the experiment. Adding experiments on larger datasets would be beneficial.

---

> ### Author Response · Authors · 2024-10-31
> **Response to the reviewer's comments (part 1)**
>
> We would like to thank the reviewer for his/her helpful comments and suggestions.
>
> Weaknesses 1,2: The experiments in Tables 1 and 2 are not fair to me. For example, JSGNN is a combination of GCN and HGCN, so the number of parameters will roughly double compared to GCN (correct me if I am wrong here). If this is the case, the results put the proposed method in an unfair position. If the above is true, the marginal improvement in Table 1 can be ignored and is not promising at all.
>
> Response: We appreciate the reviewer's observation regarding the parameter count of our model. As expected, our hybrid model requires more parameters than the base models. This is a reasonable trade-off, as our model is designed to perform well across datasets with varying geometric characteristics. As such, we think comparing the performance and model size against other benchmark "hybrid models" is more important. From our results in Table 1 - Table 3, our model has the overall best performance, particularly compared with other hybrid models ($\kappa$-GCN and GIL). In the table below, we compare the estimated number of parameters. We see that our model size is approximately the same as that of the other hybrid models. In view of the performance gain, our approach is a worthy hybrid model to consider.
> | Method             | Cora  | Citeseer | Pubmed | CS      | Photo |
> |--------------------|-------|----------|--------|---------|-------|
> | JSGNN (ours)         | 47128 | 119696   | 17720  | 219967  | 25928 |
> | $\kappa$-GCN (hybrid)             | 46112 | 118720   | 16128  | 218272  | 24128 |
> | GIL  (hybrid)              | 46700 | 119286   | 19155  | 219036  | 25380 |
> | GAT                | 23452 | 59752    | 8843   | 109244  | 12144 |
> | GCN                | 23335 | 59638    | 8339   | 109151  | 12072 |
> | HGNN               | 23335 | 59638    | 8339   | 109151  | 12072 |
> | HGCN               | 23336 | 59638    | 8339   | 109151  | 12072 |
> | HGAT               | 23431 | 59734    | 8435   | 109199  | 12120 |
>
> Weaknesses 3:  While the motivating example in Figure 2 makes sense to me, this paper does not show that it is reflected in real graphs (despite the distributions of geometric hyperbolicity in Figure 2). This paper should also plot real graphs that include tree and lattice structures.
>
> Response: Thank you for the suggestion. In the revision, we shall show (part) of the graphs of the Disease and Photo datasets. They are mostly hyperbolic and Euclidean respectively.
>
> Weaknesses 4: Figure 3 is of low quality and not professional to me.
>
> Response: Thank you for the suggestion. In the revision, we shall redraw the figure with higher resolution and clearer details.
>
> Weaknesses 5: As this paper claims that node representation is geometric (Gromov) and model-based, to determine the preferred embedding space for each node, the visualization of the node embeddings is a must-have experiment to show the learned node representations and compare them to hyperbolic and Euclidean representations.
>
> Response: Thank you for the suggestion. In the revision, we will include a discussion on the output feature representations of our model, both numerically and in terms of t-SNE visualizations.

---

> ### Author Response · Authors · 2024-10-31
> **Response to the reviewer's comments (part 2)**
>
> Weaknesses 6: Based on Weakness 1, this paper should include a discussion on the scalability of JSGNN and analyze the time complexity of JSGNN, particularly regarding computational costs compared to GNN and HGNN. Providing empirical results on how well the method scales with graph size would also be helpful.
>
> Response: For training of the model, we usually use two base models $M_{e}$ (e.g., GCN) and $M_{h}$ (e.g., HGCN) for handling Euclidean and hyperbolic structures respectively. The complexity of the message passing in our model is $O(C_e+C_h)$, where $C_e$ and $C_h$ are the message passing complexities of $M_e$ and $M_h$ respectively. Therefore, for training, the complexity of our model is in the same order (in $O(\cdot)$) as the higher one between $M_e$ and $M_h$. This means that so long as both $M_e$ and $M_h$ scale well w.r.t. graph size, so will JSGNN.  In the table below, we show the empirical run-time comparison: Time taken (in seconds) training each model and the "ratio" between the run-time of JSGNN and that of HGCN plus GCN.
> | Method   | Cora | Citeseer | Pubmed |
> |----------|------|----------|--------|
> | GCN      | 1.08 | 1.07     | 1.10   |
> | HGCN     | 3.87 | 3.96     | 4.44   |
> | JSGNN    | 18.53| 21.60    | 24.73  |
> | *Ratio*  | 3.74 | 4.29     | 4.45   |
> We see that the run-time ratio between JSGNN and HGCN plus GCN remains stable for graphs of different sizes. Therefore, the scalability of the well-studied models GCN, HGCN implies that of JSGNN. In the revision, we shall include the above discussions.
>
> Weaknesses 7: I would also recommend expanding the experiments to include synthetic graph datasets like in Figure 1, especially where both Euclidean and hyperbolic structures are observed. This would help demonstrate the effectiveness of the proposed method.
>
> Response: In fact, datasets such as Cora and Citeseer already exhibit structures characterized by Figure 1. We have analyzed both datasets and found the common features: there is a giant Euclidean component ($U_G$) and tree-like structures ($T_G$) are attached to it. Information is shown in the table below (In the table, we show the sizes of the Euclidean component $U_G$ and the tree-like components $T_G$ (in $\\%$ of the size of $G$). We also show the average geometric hyperbolicity of nodes within either $U_G$ and $T_G$, which confirms their respective structural properties).
> | Method   | Size of $U_G$ | Average $\delta_v$ | Size of $T_G$ | Average $\delta_v$ |
> |----------|-------------------|------------------------|--------------------|-------------------------|
> | Cora     | 73.9%            | 0.853                 | 26.1%             | 0.073                   |
> | Citeseer | 37.2%            | 0.872                 | 62.8%             | 0.081                   |
>
> Therefore, our results in the paper demonstrate the usefulness of JSGNN on this type of graphs. We shall include the above information and discussions when we revise the paper.
>
> Weaknesses 8: The datasets used in the experiments are quite small at this time. The largest graph is Pubmed, which may limit the scope of the experiment. Adding experiments on larger datasets would be beneficial.
>
> Response: We are sorry that we do not have enough resources to test our approach on very large graphs. Therefore, we do not have a definite answer to how JSGNN will perform on these graphs.
>
> However, the datasets being studied in our paper are diverse enough in terms of their "geometric properties". All datasets and tasks in [a], [b], and [c] are included in our paper (in fact, we have included more).  We have theoretically analyzed the complexity of JSGNN, which depends on the complexities of the hyperbolic and Euclidean base models (see Weaknesses 6). Therefore, its scalability also depends on that of the based models. For GCN, GAT, HGCN, HGAT, they scale well with graph size. Notice that message passing is a local operation, and hence model performance on graphs of medium sizes should be illustrative enough.
>
> [a] Gregor Bachmann, Gary Becigneul, and Octavian-Eugen Ganea. Constant curvature graph convolutional networks. In Proceedings of the 7th International Conference on Learning Representations, 2019.
>
> [b] Shichao Zhu, Shirui Pan, Chuan Zhou, Jia Wu, Yanan Cao, and Bin Wang. Graph geometry interaction learning. In Advances in Neural Information Processing Systems, 2020.
>
> [c] Jiaxu Liu, Xinping Yi, and Xiaowei Huang. Deephgcn: Toward deeper hyperbolic graph convolutional networks. IEEE Transactions on Artificial Intelligence, pp. 1–14, 2024

---

### Decision · Action_Editor_jvhE · 2024-11-25

**Recommendation:** Accept with minor revision

**Comment:**

I suggest the authors follow the reviewers’ suggestions and revise the draft. The following is the list of suggestions (which may not be comprehensive):

- Add statistics and evaluation results raised in the discussions with the reviewers. More specifically:
  - The distribution of the hyperbolicity values of datasets shown [here](https://openreview.net/forum?id=tHteJFeN1y¬eId=DgJcS1jEUL).
  - The comparison of computation times shown [here](https://openreview.net/forum?id=tHteJFeN1y¬eId=s7N9j8A9Rf) and [here](https://openreview.net/forum?id=tHteJFeN1y¬eId=gxm2goCxVA).
  - The comparison of parameter size of models shown [here](https://openreview.net/forum?id=tHteJFeN1y¬eId=63OfxFh4W7)
- Reflect discussions with the reviewers on the manuscript. More specifically:
  - Discussions of the computational complexity of JSGNN in terms of time and memory shown, e.g., [here](https://openreview.net/forum?id=tHteJFeN1y¬eId=RBi8nScNQb)
  - Discussions of the prediction performance on graphs with either hyperbolic or Euclidean properties shown, e.g. [here](https://openreview.net/forum?id=tHteJFeN1y)
- Fix the format issues pointed out by the reviewer. More specifically:
  - Replace Figure 3 with a clearer image.
  - Fix the format of citations.
- Discuss the limitations of this paper, for example, evaluations on larger graphs.
- Add a discussion of broader impact, as described in [Submission Guidelines and Editorial Policies](https://jmlr.org/tmlr/editorial-policies.html)

**Audience:**

GNN models that learn graph representations by utilizing graphs' hyperbolicity are one of the main topics in GNN research, starting with Chami et al. (2019) and Liu et al. (2019). In particular, the development of GNNs applicable to graphs with diverse hyperbolicity, such as Euclidean and hyperbolic, is an active area of research. This paper, which proposes a GNN model for such problems, is of interest to some audiences in the TMLR community.

**Claims And Evidence:**

This paper proposes JSGNN, a new model for graph learning tasks with hyperbolic and Euclidean geometries. It claims that the proposed model has Euclidean and hyperbolic attention mechanisms and adaptively learns a mixture of hyperbolic and Euclidean node representations according to the graph's geometry. The proposed model is claimed to achieve comparable prediction performance to existing models in the node and link prediction task.

Two reviewers found that the claims are supported by evidence.
The other reviewer raised questions about the claims and evidence. However, the authors' responses adequately addressed the reviewer's concerns. Specifically, the reviewer commented that the paper should include experiments on synthetic graphs. Experiments on synthetic graphs are preferable to see the relationship between hyperbolicity and prediction performance in ideal situations. However, these have been verified using real datasets for the Disease (hyperbolic), photo (Euclidian), and other datasets (mixture) datasets.
Also, the reviewer commented that they should add numerical experiments on large graphs. Of course, evaluating the proposed model using large graphs is preferable, and I do not prevent the authors from conducting experiments. The current experiments are sufficiently diverse in terms of hyperbolicity. I suggest considering the size of the graph datasets as a limitation of this study and leaving it for future work. Based on the discussions above, these aspects are not a critical problem of the claim and evidence.

In conclusion, this paper meets the criterion of the claim and evidence, providing appropriate foundations for the conclusion.